# Identification of risk genes for Alzheimer's disease by gene embedding

## Graphical abstract

## Authors

Yashwanth Lagisetty,
Thomas Bourquard, Ismael Al-Ramahi, ...,
Juan Botas, Kwanghyuk Lee,
Olivier Lichtarge

## Correspondence

lichtarg@bcm.edu

## In brief

Lagisetty et al. develop GeneEMBED, a method to evaluate disease-gene associations by investigating differentially perturbed interactions in a molecular network. They apply GeneEMBED on two Alzheimer's disease cohorts and three networks to identify novel candidate genes. Modulation of candidates *in vivo* showed altered neurodegeneration. They anticipate broad applicability of GeneEMBED in many complex diseases.

## Highlights

- GeneEMBED combines cohort exomic data with molecular network information

- GeneEMBED identifies 143 high-confidence candidate genes in two Alzheimer's cohorts

- Candidate genes are dysregulated in bulk brain tissue and single-cell RNA expression

- Modulation of candidate genes in animal models altered neurodegeneration

 Lagisetty et al., 2022, Cell Genomics 2, 100162
September 14, 2022 © 2022 The Authors.

# Cell Genomics

CellPress

## Article

# Identification of risk genes for Alzheimer's disease by gene embedding

Yashwanth Lagisetty,[1,2] Thomas Bourquard,[2] Ismael Al-Ramahi,[2,3,4] Carl Grant Mangleburg,[2] Samantha Mota,[2] Shirin Soleimani,[2] Joshua M. Shulman,[2,3,4,5,6] Juan Botas,[2,3,4] Kwanghyuk Lee,[2] and Olivier Lichtarge[2,4,7,8,*]

[1]Department of Biology and Pharmacology, UTHealth McGovern Medical School, Houston, TX 77030, USA
[2]Department of Molecular and Human Genetics, Baylor College of Medicine, Houston, TX 77030, USA
[3]Jan and Dan Duncan Neurological Research Institute, Texas Children's Hospital, Houston, TX 77030, USA
[4]Center for Alzheimer's and Neurodegenerative Diseases, Baylor College of Medicine, Houston, TX 77030, USA
[5]Department of Neurology, Baylor College of Medicine, Houston, TX 77030, USA
[6]Department of Neuroscience, Baylor College of Medicine, Houston, TX 77030, USA
[7]Computational and Integrative Biomedical Research Center, Baylor College of Medicine, Houston, TX 77030, USA
[8]Lead contact
*Correspondence: lichtarg@bcm.edu

## SUMMARY

Most disease-gene association methods do not account for gene-gene interactions, even though these play a crucial role in complex, polygenic diseases like Alzheimer's disease (AD). To discover new genes whose interactions may contribute to pathology, we introduce GeneEMBED. This approach compares the functional perturbations induced in gene interaction network neighborhoods by coding variants from disease versus healthy subjects. In two independent AD cohorts of 5,169 exomes and 969 genomes, GeneEMBED identified novel candidates. These genes were differentially expressed in *post mortem* AD brains and modulated neurological phenotypes in mice. Four that were differentially overexpressed and modified neurodegeneration *in vivo* are PLEC, UTRN, TP53, and POLD1. Notably, TP53 and POLD1 are involved in DNA break repair and inhibited by approved drugs. While these data show proof of concept in AD, GeneEMBED is a general approach that should be broadly applicable to identify genes relevant to risk mechanisms and therapy of other complex diseases.

## INTRODUCTION

Alzheimer's disease (AD) is a neurodegenerative disorder characterized by progressive memory loss, language deficits, and behavioral abnormalities.[1] An estimated six million individuals in the United States are afflicted with AD, and this number is projected to double by 2050.[2] The polygenic nature of AD presents an obstacle to early diagnosis and risk prediction. In late-onset AD (LOAD), the estimated genetic heritability is 60%–80%.[3,4] Though genome-wide association studies (GWASs) have identified >40 LOAD loci,[5–9] they account for only a fraction (~33%) of the heritability.[10,11] While there are many explanations for this "missing heritability" problem,[12–14] which is seen across complex diseases,[15] an attractive hypothesis suggests that genetic interactions may be a culprit.[16]

Genetic interactions are functional interactions observed among gene variants where the resulting phenotype differs from the independent phenotype of each variant.[16,17] Thus, relatively benign mutations may combine to generate complex phenotypes. Indeed, such non-additive genetic interactions have been observed in disease[18–20] and have improved current models of the genotype-phenotype relationship.[21,22] However, genome-wide discovery of pairwise genetic interactions pre-

sents major challenges. Theoretical analysis suggests that, under reasonable assumptions, nearly 500,000 samples would be needed to identify statistically significant genetic interactions.[16] The potential use of prior knowledge to compensate for necessary sample size has motivated the development of network informed gene prioritization methods for various diseases.[23–28] These approaches do not typically use patient-specific genetic data. However, when they do, they often rely on expression data (e.g., HIT'nDRIVE)[27] or they are built for somatic mutations (e.g., HotNet2)[28] and are not immediately amenable to the case-control study designs typical of germline GWASs.

Advances in graph representation learning open new opportunities to analyze genomes in the context of biological networks. Graph learning techniques have been successful in a variety of biological applications, including predicting protein-protein interactions[29–33] and drug responses or side effects.[34–37] Specifically, node embedding enables machine learning on networks by compacting the qualitative and quantitative properties of a network node in a mathematically suitable framework. For example, Deep Walk[38] and Node2Vec[39] use random walk algorithms to represent nodes as vectors. Alternatively, Graph Convolutional Networks[40] or Graph Attention Networks[41] use graph neural network architectures to construct node representations

instead. Regardless of the approach, node embeddings should conserve the relative properties between original graph nodes, meaning that similar nodes should embed similarly. We hypothesize, based on this principle, that differences in a gene's embedding in a disease network compared with its embedding in a healthy network may reflect a role in disease pathology.

This motivated us to develop GeneEMBED (gene-embedding-based evaluation of disease-gene relevance) to pinpoint genetic risk factors of disease by examining the differential perturbation patterns of gene interactions. The approach takes a predefined molecular network and annotates it with the functional impact of protein coding variants across cases and separately controls. Importantly, the approach considers all protein coding variants in estimating gene-level perturbed protein function. Machine learning performs embeddings on each network and then finds which genes have the most difference in case versus controls embeddings. Notably, this approach addresses the limitations of standard models by feasibly assessing the contribution of pairwise, and higher order, genetic interactions on disease and doing so with a case-control study design of typical genome-wide studies.

While this approach is general and applicable to many complex diseases, we tested this in two LOAD datasets: the Alzheimer's Disease Sequencing Project (ADSP) (dbGaP: phs000572.v7.p4) Discovery cohort comprising 2,729 affected (AD+) individuals and 2,440 healthy (AD−) controls and the Extension cohort with 481 AD+ and 488 AD− individuals (NIGADS: NG00067). To assess robustness of GeneEMBED, we used two variant impact scoring methods, Evolutionary Action (EA)[42] and PolyPhen2 (PPh2),[43] and we tested three different molecular interaction networks: STRING,[44] HINT,[45] and a brain-specific network.[46,47]

Candidate genes from the Discovery and Extension cohorts were consistent with one another and with known AD genes. The candidates interacted with manually curated AD-associated genes and were dysregulated in AD brains. Functional *in silico* analysis showed they were involved in pathways relevant to AD, including for cell cycle and DNA replication. *In vivo* perturbation analysis confirmed that GeneEMBED genes were modifiers of tau and β-amyloid-induced phenotypes in well-established *Drosophila* AD models,[48–50] and their modulation in mice showed abnormal neurological phenotypes, supporting their role in normal neuronal maintenance and function. Importantly, many GeneEMBED candidates are druggable with already approved compounds. Overall, these results point to new targets for therapeutic development in AD and broadly support a novel and general paradigm to interrogate other complex genetic diseases.

## RESULTS

### GeneEMBED identifies genes that are perturbed in AD
With a view to discover AD genes, GeneEMBED aims to combine the integrative features of network biology with machine learning to find genes with functional interactions perturbed differently among cases and controls, due to mutations. First, GeneEMBED builds a personalized functional impact network by calculating a perturbation score (PS) for each gene of each

subject of a cohort. This score reflects all non-synonymous variants in the gene (*v*); the impact of each variant is estimated by either EA[42] or PPh2[43] (Variant Impact Score$^{EA}$ and VIS$^{PPh2}$, respectively) and zygosity (*zyg*) (Figure 1A; STAR Methods). The PS scores are then mapped to a gene network of choice, such as the STRING protein-protein interaction network, by setting the weight of an existing edge between two genes as the sum of their PS score. Finally, the edge weights are averaged across all cases, or separately across all controls, to produce two global cohort networks that compile the aggregate mutational perturbations of protein-protein interactions in cases and in controls. Both networks are then processed with the GraphWave[51] machine learning algorithm, which applies an unsupervised diffusion-aided wavelet decomposition to assign a continuous vector-valued embedding to each gene or node. This embedding is based on the topological (geometric distribution of the edges in the node's vicinity) and functional (functional information associated with each edge) properties surrounding the gene in the network. As a result, the vector assigned to each gene represents the integrated functional perturbation of the variants in its network neighborhood. The final step applies principal-component analysis (PCA) to identify vectors with significant differences between the case and control networks (false discovery rate [FDR] < 0.01), suggesting distinct perturbation patterns in these genes between AD versus controls.

Next, to test the algorithm and identify genetic factors underlying AD, we applied GeneEMBED to the whole-exome sequencing (WES) and whole-genome sequencing (WGS) data from the ADSP Discovery and Extension cohorts, respectively, using either $VIS^{EA}$ or $VIS^{PPh2}$ for the variant impact score and initially the STRING protein-protein interaction network. In addition, we applied GeneEMBED to healthy control versus healthy control using both $VIS^{EA}$ and $VIS^{PPh2}$ to identify potential false-positive (FP) genes. After removal of FPs, GeneEMBED identified 69 AD− candidates in the Discovery Cohort and 119 candidates in the Extension cohort with $VIS^{EA}$ and 128 candidates in the Discovery Cohort and 120 genes in the Extension cohort (Table S1) with $VIS^{PPh2}$.

Fourteen genes overlapped between the Discovery and Extension cohorts when using $VIS^{EA}$ (one-tailed hypergeometric p ≈ 1.86e−16). Of these, nine genes had evidence in literature documenting their association with AD (Figure 1B; *APOE*, *CSF1R*, *ILR4*, *MAPK6*, *MAPT*, *REST*, *RIPK4*, *SP3*, and *TRIB3*).[52–60] Particularly notable were *MAPT* and *APOE.* Neurofibrillary tangles, one of the primary AD biomarkers, are aggregates of hyperphosphorylated *MAPT* gene products.[61] *APOE*, on the other hand, is one of the strongest genetic predictors of AD.[61]

Similarly, 16 genes overlapped between Discovery and Extension cohorts when using $VIS^{PPh2}$ (one-tailed hypergeometric p ≈ 4.25e−15), of which six have been previously linked to AD pathology (Figure 1B; *CCT5*, *ERBB2*, *MAPK6*, *REST*, *SYNJ1*, and *TP53*).[55,57,62–65] GeneEMBED-$VIS^{PPh2}$ did not recover *APOE* in the Discovery cohort but did so in the Extension.

GeneEMBED also identified well-known genes in which rare variants are associated with AD, including *TREM2*[66] and *SORL1*,[67] though these genes are recovered only in the Discovery cohort. Comparing $VIS^{EA}$ to $VIS^{PPh2}$, 34 genes overlapped in the Discovery cohort (one-tailed hypergeometric p ≈ 1.46e−53)

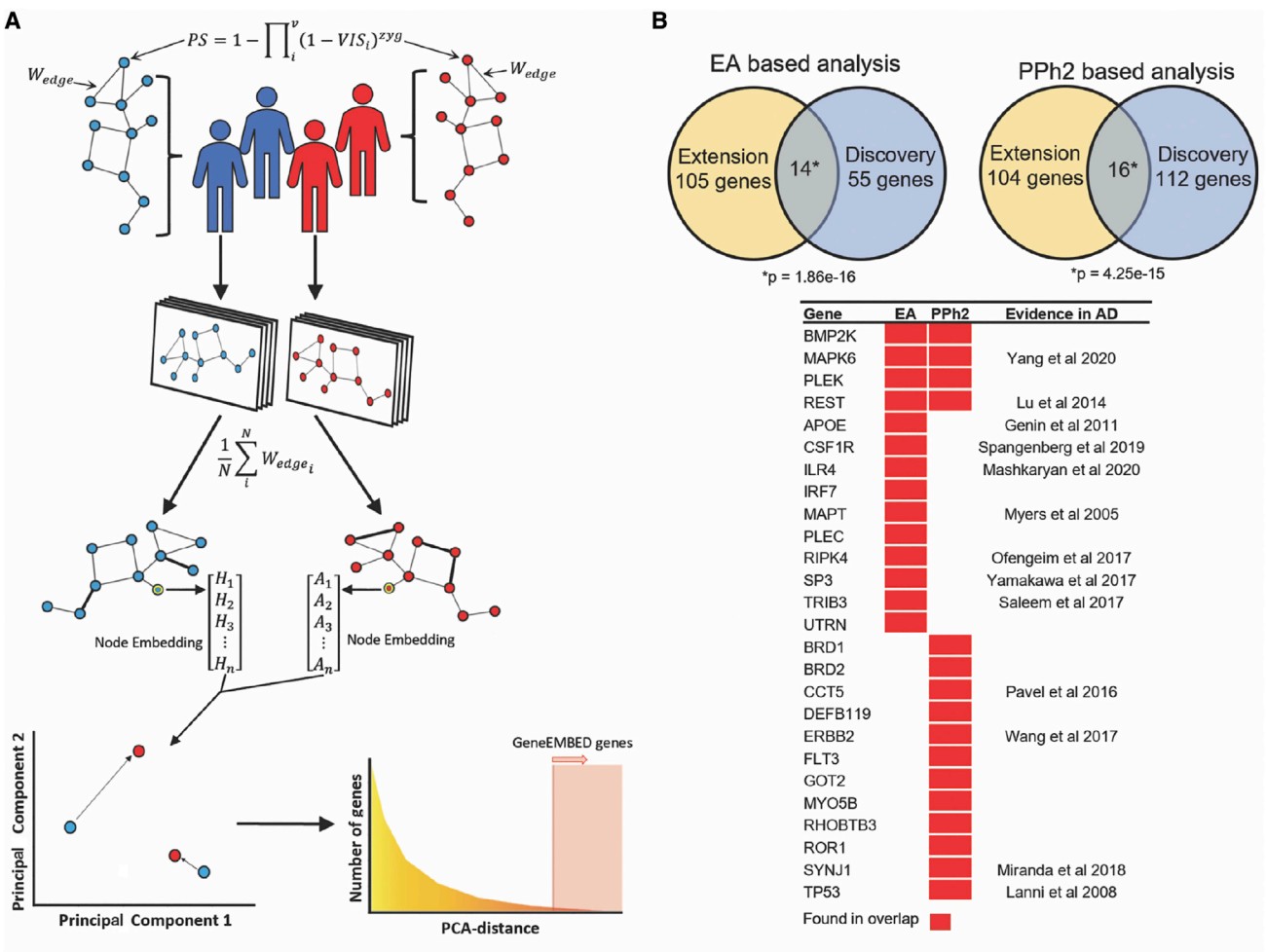

**Figure 1. Overview of GeneEMBED and AD candidate genes**

(A) GeneEMBED: for an individual, genes are first assigned a perturbation score (PS) consolidating information from all the gene's variants appearing in the individual. The gene PS estimates the total loss of function probability given various combinations of variant level loss-of-function probabilities. Edge weights for an individual's network are calculated by the sum of the PS of the connected genes. Edge weights are then averaged over to construct one case specific and one control specific graph. Node embedding is performed on the genes in the two networks. Finally, embeddings are projected in a PCA space to measure distances between nodes in case and control networks.

(B) GeneEMBED using EA identified 69 candidate genes in Discovery and 119 in Extension with 14 overlapping genes, significant by one-tailed hypergeometric test. In PPh2 analyses, 128 candidate genes were found in Discovery and 120 in Extension with 16 overlapping genes, significant by one-tailed hypergeometric test. A large portion of overlapping genes have been previously implicated in AD biology.

and 44 genes overlapped in the Extension network (one-tailed hypergeometric p $\approx$ 2.46e-64), indicating concordance between these two impact scores. Lastly, we found that four genes overlapped among all cohort-VIS combinations with a one-tailed hypergeometric p $\approx$ 8.58e−10. These data suggest that GeneEMBED is robust to inter-cohort variability as well as differences in impact scoring systems and can recover several well-characterized, positive control AD genes.

In order to control against a standard method for inferring gene-disease associations, we used Multi-marker Analysis of GenoMic Annotation (MAGMA), which prioritizes genes based on multiple regression analysis. This method can be performed genome-wide, allowing it to be used for gene discovery.[68] MAGMA identified 31 AD-associated genes in the Discovery

cohort and only seven in the Extension, with no overlap (Table S2). MAGMA in the Discovery cohort shared only *APOE* with both GeneEMBED-*VIS^EA* analyses and GeneEMBED-*VIS^PPh2* in Extension while overlapping with *VIS^PPh2* analysis in Discovery cohort by two genes *SORL1* and *PRIM1*. Similarly, MAGMA in Extension only shared *TPO* with *VIS^PPh2* in Discovery and did not overlap with any other analyses. Of the 31 MAGMA candidates from the Discovery cohort, nine had been previously associated with AD, including *APOE* and *TOMM40*.[52,69] This indicates that MAGMA was less effective and less reproducible at this small sample size.

To assess the recovery of GeneEMBED in a systematic manner, we measured one-tailed hypergeometric overlaps between GeneEMBED candidates and 208 AD-associated genes

in the DisGeNet database. DisGeNet compiles gene-disease associations based on genetic, clinical, and animal model curation.[70] We found a significant overlap (p = 0.012–5.3e−4) between GeneEMBED candidates and DisGeNet genes across functional mutational impact methods ($VIS^{EA}$ versus $VIS^{PPh2}$) and cohorts (Discovery versus Extension; Table S3). In addition, we found significant overlaps between AD-associated genes from the comparative toxicogenomic database (CTD)[71] and GeneEMBED-$VIS^{EA}$ in both the Discovery (p = 0.047) and Extension (p = 3.3e−3) cohorts. MAGMA candidates in the Discovery cohort recovered similar significant overlaps (Table S3; the low number of MAGMA candidates in the Extension cohort prevented a similar analysis).

These data suggest that GeneEMBED is able to significantly recover several known AD genes despite large differences in cohort sizes. Moreover, MAGMA was unable to reproducibly retrieve genes between the Discovery and Extension cohorts, while GeneEMBED found significant overlaps. Taken together, these findings demonstrate the robustness of GeneEMBED, compared with MAGMA, to both inter-cohort variability and sample size. Overall, GeneEMBED identifies candidates distinct from MAGMA, which are nonetheless enriched for known AD-associated genes, suggesting an identification of disease-relevant signal.

### GeneEMBED candidates are robustly connected and relevant to AD

To assess the role of GeneEMBED candidates, we asked whether they are implicated in molecular changes related to AD, specifically, dysregulated gene expression as tallied by the Accelerating Medicines Partnership Alzheimer's Disease (AMP-AD) RNA sequencing from seven brain regions.[72–77] To focus on novel genes, we removed GeneEMBED genes that overlapped with any of five curated AD gene sets (DisGeNet, CTD, ClinVar, GWAS Meta 1, and GWAS Meta 2).[5,6,70,71,78] The remainder was significantly dysregulated in the temporal cortex of AD patients (TCX) (one-tailed hypergeometric p < 0.05; Figure 2A), independent of both the functional impact method ($VIS^{EA}$ versus $VIS^{PPh2}$) and the cohort (Discovery versus Extension). However, GeneEMBED-$VIS^{EA}$ candidates were also dysregulated in the parahippocampal gyrus (PHG) (Figure 2A) for both cohorts and in the cerebellum (CBE), frontal pole (FP), superior temporal gyrus (STG), and dorsolateral prefrontal cortex (DLPFC) (one-tailed hypergeometric p < 0.05; Figure 2A) for the Extension cohort, whereas that was only true for GeneEMBED-$VIS^{PPh2}$ on the CBE, also in the Extension cohort. MAGMA, in contrast, found no enrichment in dysregulated genes.

In addition to this, the number of brain regions with significant dysregulation of candidate genes for GeneEMBED-$VIS^{EA}$ in the Discovery cohort and GeneEMBED-$VIS^{PPh2}$ in the Extension cohort was on par with the number from two AD GWAS meta-analyses (permutation-based one-tailed Z test p ≈ 1.2e−2, p ≈ 2.3e−3, pGWAS Meta 1–2.8e−3, and pGWAS Meta 2–4.9e−3, respectively). Remarkably, GeneEMBED-$VIS^{EA}$ applied to the Extension cohort identified candidates significantly dysregulated in six brain regions in AD (p ≈ 1.6e−13) (Figure 2B). Moreover, many of these genes were also dysregulated in single cells (Figure 2C). Together, these data indicate a

strong link between this group of candidate genes and AD pathology. This link, however, could be either causative or responsive.

Next, we tested whether novel GeneEMBED candidates were connected to AD-reference gene sets. For this, we measured how well information propagated between them and AD-associated genes in a protein-protein interaction (PPI) network[79–81] using the nDiffusion method.[82] Area under a receiver-operator curve (AUROC) measures the strength of their interaction, and a permutation-based Z score over 100 permutations is calculated for the significance of the observed AUROC compared with a distribution of random gene sets. We used two disease-gene association databases (DisGeNet—208 genes[70] and CTD—103 genes[71]) and three variant-based reference gene sets for AD (GWAS Meta 1–25 genes,[5] GWAS Meta 2–38 genes,[6] and ClinVar 21 genes[78]). The GeneEMBED candidates showed statistically significant diffusion (ROC > 0.5 + Z score > 2) to most selected AD-associated gene sets, regardless of the cohort (Figure 3; Table S4; AUROC = 0.63–0.84; Z = 2.03–5.64). Interestingly, MAGMA candidates also diffused significantly to DisGeNet, CTD, and ClinVar, but not to the two GWAS datasets. These data suggest that the GeneEMBED candidates are functionally and significantly connected to previously curated AD-associated genes, further suggesting an identification of disease-relevant signal.

To test GeneEMBED's utility and robustness in alternate PPI networks, we replicated the experiments from the above sections using the HINT network[45] of curated high-quality PPIs and a second network of physical PPIs specific to brain tissue.[46,47] First, using the HINT network, only $VIS^{EA}$ in Discovery showed significant recall of genes from the CTD, GWAS Meta 1, and GWAS Meta two references (one-tailed hypergeometric p = 0.0014, 0.0058, and 0.015) (Tables S5 and S6). However, nDiffusion found both $VIS^{EA}$ and $VIS^{PPh2}$ in Disc were significantly connected to all curated gene sets except GWAS Meta 1, with AUROCs = 0.62–0.77 (permutation-based Z score = 2.31–5.77) and AUROCs = 0.62–0.76 (permutation-based Z score = 2.6–3.89) (Table S7), respectively. $VIS^{EA}$ and $VIS^{PPh2}$ in Extension also had significant network connectivity with CTD and DisGeNet gene lists with AUROCs = 0.75 and 0.7 (Z = 3.33 and 5.16) and AUROCs = 0.74 and 0.67 (permutation-based Z score = 3.32 and 4.91).

Alternately, using the brain-specific PPI, both $VIS^{EA}$ and $VIS^{PPh2}$ in Discovery had significant interactions to the curated gene sets, with AUROCs = 0.63–0.78 (permutation-based Z score = 2.11–3.91) and AUROCs = 0.64–0.82 (permutation-based Z score = 2.43–6.07) (Tables S8, S9, and S10). $VIS^{EA}$ in Extension found significant relatedness to CTD and DisGeNet with AUROCs = 0.77 and 0.69 (permutation-based Z score = 4.64 and 5.07). $VIS^{PPh2}$ in Extension did not show any significant links to the curated gene sets. These data show that GeneEMBED robustly identifies genes enriched for functional interactions to curated sets of AD-related genes using a variety of alternative PPI networks. Interestingly, a large number of genes were repeatedly identified among two or more GeneEMBED analyses across cohorts, VIS systems, and PPI networks (Figures S1 and S2; Table S11), suggesting a potential role in AD.

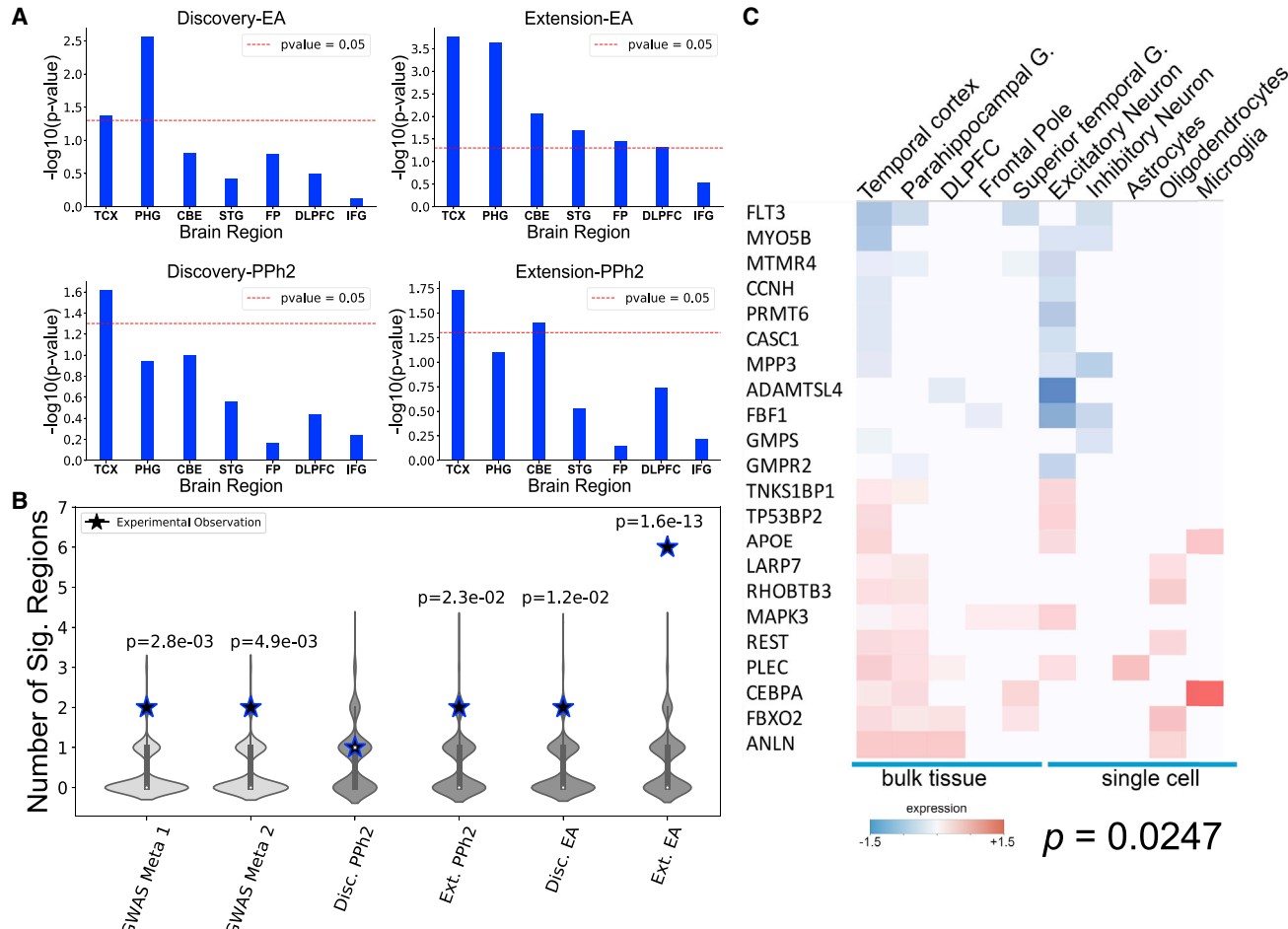

**Figure 2. GeneEMBED candidates are differentially expressed in AD brain tissue**
(A) One-tailed hypergeometric enrichment of GeneEMBED candidates against differentially expressed genes from seven brain regions: cerebellum (CBE), temporal cortex (TCX), frontal pole (FP), inferior frontal gyrus (IFG), parahippocampal gyrus (PHG), superior temporal cortex (STG), and dorsolateral prefrontal cortex (DLPFC).
(B) Comparison of RNA-sequencing-based enrichment between known AD gene sets and GeneEMBED candidates. Stars indicate the number of brain regions with significant enrichment in each gene set by permutation testing. Violin plot shows the distribution of expected number of enriched brain regions when using random gene sets.
(C) Among the 143 high-confidence genes, a significant number (22; one-tailed Fisher's exact test; p = 0.0247) showed differential expression in both bulk tissue from various brain regions and in single-cell sequencing of neuronal cell types.

## GeneEMBED candidates are functionally connected and enriched for *in vivo* modulators of neuronal dysfunction triggered by tau and β-amyloid

The significant overlap in GeneEMBED candidate genes observed across cohorts and networks (Figure S2) indicates that GeneEMBED may be identifying specific pathways where an increased concentration of mutational load modulates AD risk. To investigate this, we performed functional enrichment analysis. We constructed a network in STRING with 143 high-confidence hits. These genes were selected using the criteria that they must have been identified at least twice in the same network either across cohorts or across *VIS* methods. Genes were prioritized based on the degree of overlap across networks with more recurrent genes ranking higher, provided that they were never identified in any of the healthy control versus healthy control assays (Figure S1; Table S11).

Interestingly, this network showed significant PPI enrichment (STRING PPI enrichment p = 9.56e−07). After clustering with a Louvain algorithm, 127 of the 143 candidate genes mapped to significantly enriched pathways (Figure 4), including, among others, (1) mechanisms involved in glial biology (glial-cell-derived neurotrophic factor receptor);[83,84] (2) inflammation (regulation of IP-10 production, positive regulation of transforming growth factor β1 (TGFβ1) production, and chemokine signaling), which is known to be dysregulated in AD;[61] (3) clearance of protein aggregates (regulation of aggrephagy and MTOR signaling); and (4) extracellular signaling cascades. These cascades involved Wnt/β-catenin, G-alpha, or ErbB, which are dysregulated in AD[63,85] and modulate neurodegeneration in animal models,[86] or Syndecan-3, which may play a role in tau and β-amyloid internalization.[87] (5) The largest functional module among the high-confidence GeneEMBED candidates is related to DNA

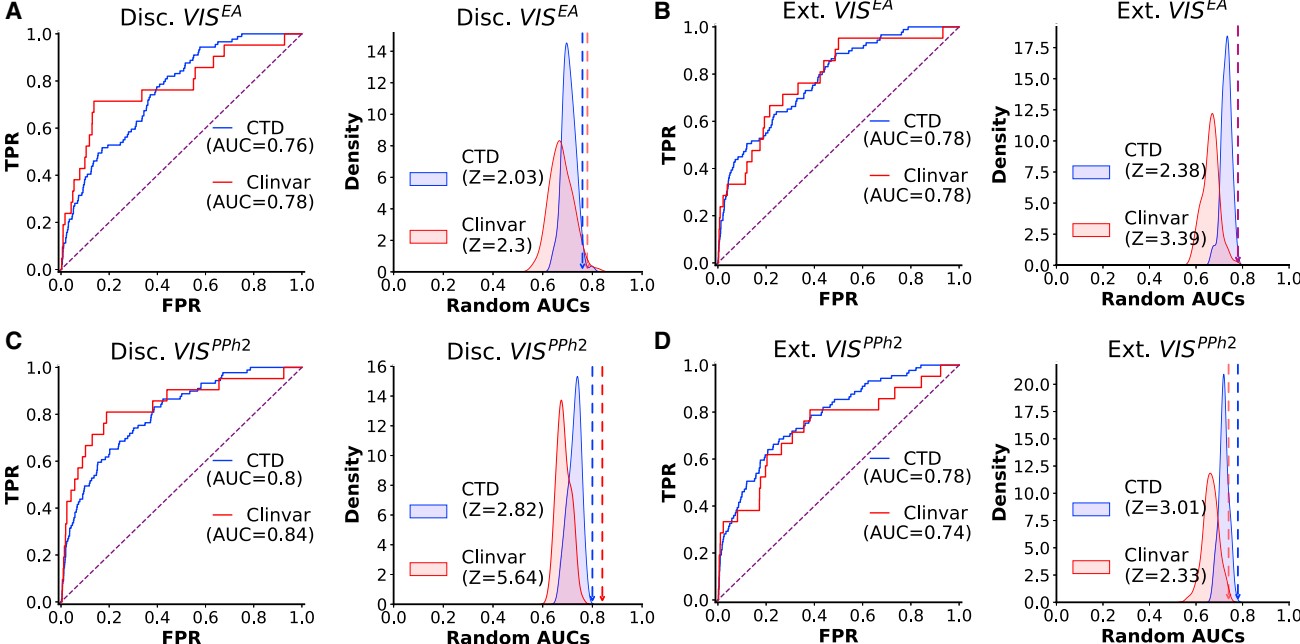

**Figure 3. GeneEMBED candidates are significantly related to curated sets of AD genes**
(A) Receiver operator characteristic curves are shown for Disc. VIS[EA] for network diffusion to CTD and ClinVar AD gene sets. To determine significance of observed area under the curve (AUC), a permutation testing strategy is used wherein random gene sets of the same size are generated 100 times and analyzed through nDiffusion to create a random distribution of AUCs. Reported Z-scores are calculated relative to these backgrounds. y axis of the ROC plots are true positive rates (TPRs), and x axis is false-positive rate (FPR). Similarly, y axis of the Z score distribution is probability density, and x axis is the AUROC score of random gene sets.
(B–D) Analogous plots are shown for (B) Ext VIS[EA], (C) Disc VIS[PPh2], and (D) Ext VIS[PPh2].

double-strand break repair. Interestingly, genes involved in double-strand break repair regulation modulate neurodegeneration in animal models,[59] and others involved in DNA quality control accumulate in AD brains.

These pathways suggest that modulating GeneEMBED genes may impact neuronal function. This hypothesis is supported by the fact that the 143 high-confidence hits are enriched in differentially expressed genes both in bulk and in single-cell transcriptomic datasets from AD *post mortem* brains (one-tailed Fisher's exact test p = 0.0247; Figures 2C and 4).

While many genes have been investigated in AD mouse models to understand their contribution to disease, it is currently impractical to perform this type of analysis with large gene collections. To circumvent this limitation and systematically measure whether GeneEMBED candidates play important roles in CNS, we asked whether modulation of their mouse homologs would cause any neurological phenotypes as tallied in the Mouse Genome Informatics (MGI) database.[88] This would reveal whether gene candidates are involved in neuronal maintenance and function and whether their loss of function may constitute a risk factor for AD or be a trigger for neurodegeneration.

We found that, out of 139 high-confidence genes with homologs, 48 (35%) showed abnormal nervous system phenotypes (one-tailed Fisher's exact test p = 0.00024) when modulated. Notably, among these, a subset of 25 mouse homologs also showed abnormal behavioral and neurological phenotypes (one-tailed Fisher's exact test p = 0.049). Finally, an additional

11 homologs showed only abnormal behavioral and neurological phenotypes (Figure 4 shows genes whose modulation causes CNS-associated phenotypes in mice as red or yellow border nodes). Of note, neither the ADSP variant datasets nor the STRING or HINT networks used by GeneEMBED have any bias toward genes expressed in the brain or in neurons. Therefore, the observed enrichment in genes mediating normal neuronal function increases confidence in GeneEMBED and with the potential pathogenic or protective roles of the genes it finds.

To further ascertain the role of GeneEMBED genes in neurodegeneration, we next turned to *in vivo* experiments. Mouse models recapitulate neuronal dysfunction and neuropathological features of AD; however, they are not amenable for testing a high number of candidates using functional assays. Conversely, cultured cells fail to recapitulate core AD traits (age dependence, circuit dysfunction, and neuron-glia interplay). Therefore, to optimally validate the GeneEMBED candidates in the AD context *in vivo*, we resorted to *Drosophila* AD models, which capture important core AD traits, including age dependence and protein accumulation.[89] This approach is supported by our previous *Drosophila* work in the context of AD and other neurodegeneration disorders, where therapeutic targets identified in *Drosophila* have gone on to be validated in mouse or induced pluripotent stem cell (iPSC)-derived neuronal models.[48,49,89–94]

For the GeneEMBED candidates, we modulated the levels of their *Drosophila* homologs in two well-validated *Drosophila* AD models[48–50] to test the effect of each candidate on neuronal

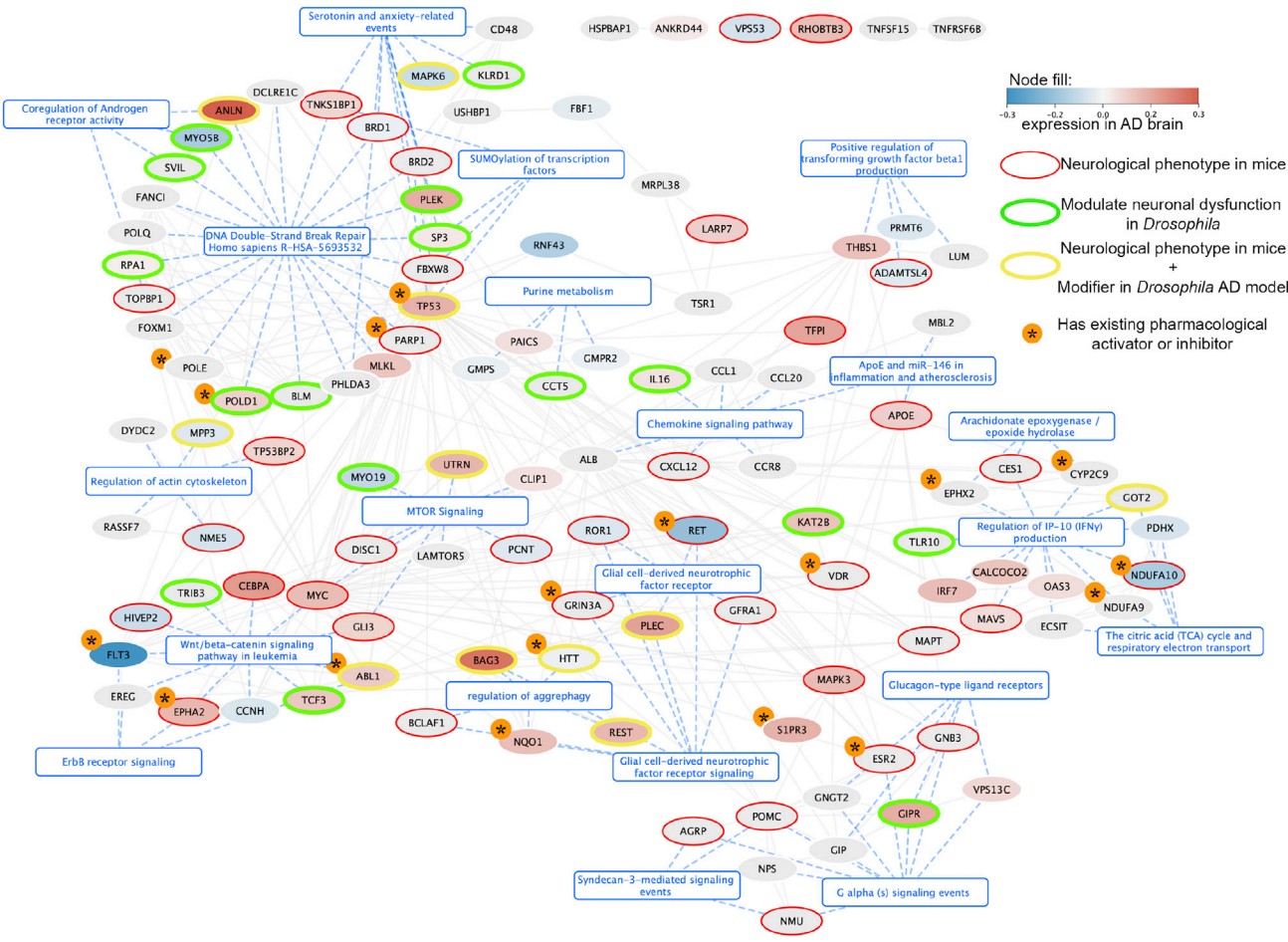

**Figure 4. Interaction network among 143 high-confidence genes**
Network is built using STRING edges. Nodes are colored based on their differential log2 fold change expression in AD brains. Red rings around genes indicate that they were reported in MGI to have abnormal neurological phenotype when knocked out. Green rings indicate that the gene was observed to modify AD phenotype in *in vivo* experiments on AD *Drosophila* models. Yellow rings indicate genes that were observed to both modify AD phenotype in *Drosophila* models and have reported abnormal neurological phenotype in knockout (KO) mouse models in MGI. Genes with asterisk next to them are those that have pre-existing FDA-approved pharmacological activator or inhibitors, indicating potential targets for drug-repurposing studies.

dysfunction caused by amyloid (secreted Aβ42) or Tau (2N4R hTau) in the CNS. Expression of secreted *β42* or human *tau* specifically in post-mitotic neurons induces progressive nervous system dysfunction in *Drosophila* that can be monitored by measuring the motor performance of the animals as they age.

First, we filtered out high-confidence candidate genes that did not have *Drosophila* homologs or available alleles in public repositories. We then tested the resulting 43 genes using both overexpression as well as loss-of-function alleles whenever possible. We found that 28 *Drosophila* genes were modifiers of the β42- and/or tau-induced neuronal dysfunction (Figures 4, green and yellow border nodes, S3, and S4). We further found that, of these 28 modifiers, five genes (UTRN, REST, PLEC, BAG3, and TP53) also showed evidence of dysregulation in human *post mortem* AD brain transcriptome and abnormal neurological phenotypes in knockout mice. Interestingly, both the MGI hits as well as the *Drosophila* modifiers are evenly distrib-

uted between the different functional clusters (Figure 4), indicating that all these pathways may potentially modulate AD pathogenesis.

Importantly, some of the *Drosophila* alleles used (inducible overexpression and short hairpin RNA [shRNA] lines) were targeted specifically to neurons and therefore likely exerted their effects specifically in neuronal cells. However, other alleles used were classical loss-of-function or classical rescue constructs (using the endogenous gene promoter); in those cases, the effect may be cell-autonomous or non-cell autonomous, for example, through modulation of important functions in glial or muscular cells. In addition, while some of the modifiers identified may exert their effect through modulating the accumulation of *tau* or *β42*, others may act by protecting or potentiating the predisposition of neurons to degenerate or even by causing certain levels of neurodegeneration themselves. A complete list of the modifier alleles as well as brief description of their putative effect on their target gene are available in Table S12.

Given the likely neurological role of these high-confidence GeneEMBED candidates, we investigated their therapeutic potential. Among the 143 genes, 21 have drugs that have been characterized as agonists or antagonists of their function (Table S13). Interestingly, of the total 109 compounds activating or inhibiting these genes, 35 have co-mentions with AD in the PubMed database.

Noteworthy among these druggable candidates are *EPHA2* and *S1PR3*, both of which were upregulated in AD brains. *EPHA2* has two inhibitors (regorafenib and dasatinib), both of which have shown neuroprotective effects in mouse AD models.[95,96] *S1PR3* has an agonist (fingolimod) that also has therapeutic benefit in mice.[97] In addition, two genes, *FLT3* and *RET*, are inhibited by sunitinib, which inhibits cerebrovascular activation to improve cognitive function mouse AD models.[98]

Among the genes whose knockdowns ameliorated neurodegeneration in *Drosophila* AD models, three (*ABL1*, *TP53*, and *POLD1*) have pharmacological agents with previously demonstrated inhibitory effects. While ABL1 inhibition is already being pursued in the context of AD,[99,100] TP53 and POLD1 remain to be explored. Together, our results demonstrate that high-confidence GeneEMBED candidates show significant enrichment in modifiers of tau and β-amyloid phenotypes in *Drosophila* models, are differentially expressed in AD brain tissue, and show abnormal neurological phenotypes when modulated in mouse models. These findings highlight the ability of GeneEMBED to successfully identify genes involved in disease pathology, some of which have significant therapeutic potential.

## DISCUSSION

AD is the leading cause of dementia worldwide. As its prevalence rises, the need to identify therapeutic targets, potential biomarkers, and risk predictive strategies is urgent. These tasks are complicated by the fact that, although several AD genes have been discovered, they only partially account for the role of genetics in the disease.[10,11] Here, we developed GeneEMBED, a new approach to pinpoint genetic risk factors of disease by examining the differential perturbation patterns of gene interactions. Though, in this study, we analyze AD as proof of concept, GeneEMBED is a general approach applicable to many complex polygenic diseases.

When applied to the ADSP cohorts, GeneEMBED identified 143 candidate genes that interacted significantly with previously known AD genes (Z score = 2.03–6.07) and were differentially expressed in bulk tissue and single cells of AD cases (p = 0.0247). While testing such a large collection of genes in AD-related mouse models is currently not possible, we sought to identify experimental links between the GeneEMBED candidates and neuronal biology.

We validated candidate genes *in vivo* using two well-characterized *Drosophila* AD models and utilized the MGI database to identify functional links between the GeneEMBED genes and neurological phenotypes. These genes were also linked to known AD pathways and revealed several novel and potentially druggable targets. These pathways included functions related to glial biology, inflammation, protein aggregate clearance, and signaling cascades.

While inflammation plays a large role in the pathogenesis of AD, our enrichments draw attention to the regulation of interferon-gamma-induced protein 10 (IP-10) production. In AD patients, IP-10 has elevated expression in astrocytes and shows positive correlation between cerebrospinal fluid (CSF) levels and cognitive impairment.[101] In AD transgenic mice, it co-localizes with amyloid plaques.[101] Interestingly, among genes responsible for enrichment in this function, three (*NDUFA10*, *GOT2*, and *TLR10*) show modulation of an abnormal phenotype in animal models (Figure 4), while another four (*NDUFA10*, *NDUFA9*, *EPHX2*, and *CYP2C9*) have approved pharmacological activators or inhibitors (Figure 4).

Functions related to glial biology highlighted glial-cell-derived neurotrophic factor (GDNF) receptor (GFRa1) signaling. Studies in transgenic AD mice found that overexpression of *GDNF* induced neuroprotective effects and improved learning and memory.[83] Restoration of *GDNF* effects by introduction of exogenous *GFRa1* into cortical AD neurons has been shown to alleviate neuronal death.[84]

Strikingly, we found that all eight genes (*RET*, *ROR1*, *GRIN3A*, *PLEC*, *GFRA1*, *BAG3*, *NQ O 1*, and *BCLAF1*) responsible for enrichment in this pathway showed modulation of abnormal neurological phenotypes in mice and *Drosophila* (Figure 4). Of these, *RET*, *GRINA3*, and *NQ O 1* all have pharmacological activators or inhibitors that are US Food and Drug Administration (FDA) approved. *GRINA3*, specifically, interacts with acamprosate, which has been associated with decreased incidence of dementia in population studies and has been seen to alleviate cognitive defects in amyloid precursor protein (APP) transgenic mice.[102,103] Further studies of these gene candidates are needed to disentangle their relationship with AD; however, they present interesting and viable targets for potential therapeutic research.

Several GeneEMBED hits represent novel and unsuspected candidates for AD. Particularly noteworthy were *PLEC* and *UTRN*, which, to our knowledge, have not been studied in AD. Both genes were repeatedly identified in multiple GeneEMBED analyses and were significantly upregulated in bulk tissue AD brains, their modulation causes abnormal neurological phenotypes in mouse models,[104,105] and they are genetic modifiers of AD-related phenotypes in *Drosophila*.

*PLEC* encodes for plectin, a cytoskeletal protein involved in intermediate filament networks and interacting with actinomycin and microtubules. Mice deficient in *PLEC* isoform *P1c* in neurons demonstrate altered pain sensation and reduced learning and long-term memory due to increased accumulation of tau proteins with microtubules.[104] Proteomic studies have also associated *PLEC* with AD pathology.[106,107] *UTRN* encodes for utrophin, another component of the cytoskeletal system. Though *UTRN* is downregulated in CA1 neurons containing neurofibrillary tangles,[108] its role in the development of tangles is still unclear. The numerous modalities in which *UTRN* and *PLEC* show associations to AD phenotype warrant deeper and more detailed studies to unravel their role in the disease.

In a similar vein, we found two additional genes with links to AD worth highlighting (*TP53* and *POLD1*)[109,110] and whose knockdown in *Drosophila* alleviated AD-related phenotypes. Moreover, both of these genes have pre-existing FDA-approved pharmacological inhibitors. We found four compounds

(clofarabine, cytarabine, fludarabine, and gemcitabine) that inhibit *POLD1* and one compound (bortezomib) that inhibits *TP53*. Given the distinct effects of these genes in animal models and their druggability, these genes would be priority candidates for further characterization and study in animal models.

As a genomic tool, GeneEMBED searches for genes that influence disease risk by considering mutational perturbations of function in their molecular interaction network. This is in contrast to variant or gene-based association methods that treat individual genes or variants as independent and isolated risk loci.[68,111–114] To evaluate the functional perturbation of a gene in a disease, GeneEMBED integrates two distinct techniques: variant impact estimators and node-embedding algorithms (Figure 1A).

The first of these, variant impact estimators, predict the probable effect of a coding mutation on protein function based on a variety of data. EA is an untrained approach that uses the evolutionary history of sequence variations and phylogenetic divergence to predict the impact of a variant. PPh2 evaluates impacts by applying machine learning tools on sequence and structure features. These estimates are combined across all variants in a gene to predict their total impact on protein function.

The second technique, node embedding, is a machine learning process that seeks to represent the complex topological properties of a node in an easily manipulatable form. By weighing the interactions of a gene with the sum of its mutational impact and those of its interactors, GeneEMBED uses the perturbed interactions of a gene as learning features rather than their singular mutational burden. Combining these features with node embedding allows GeneEMBED to estimate the differential perturbation of genes in cases versus controls, thereby identifying genes whose disease contribution would not have been apparent in single-gene analyses. For example, in AD, the single-gene approach MAGMA did not identify *NQ O 1* ($p_{MAGMA} \approx 0.33$) as disease associated despite its links to AD.[115–118] However, its differentially perturbed network interactions between cases and controls allow GeneEMBED to identify *NQ O 1* with statistical significance (Figure S5). This suggests that GeneEMBED identifies genetic processes distinct from those found by standard tools and can offer complementary insights into the factors defining complex diseases.

The integrative framework of GeneEMBED provides other advantages. First, the integration of network information allows GeneEMBED to be robust to sample sizes. In our analysis of AD, GeneEMBED was able to reliably reproduce findings from the full ADSP Discovery cohort with successively smaller subsampled cohort sizes (Figure S6A). More than that, GeneEMBED was robust to variations between different cohorts, recovering significant overlaps (p = 1.86e−16 and 4.25e−15) in genes identified in the ADSP Discovery and Extension datasets, a challenging task for standard prioritization tools at these sample sizes. Nevertheless, in order to optimally account for the various factors leading to inter-cohort variability and increase robustness of findings, we recommend readers to validate potential candidate gene lists across two or more cohorts.

This framework is also flexible in that it is compatible with many different variant impact estimators. Here, we used EA due to its consistently good performance in blind, objective

studies[119,120] and overall utility in genomic studies,[121,122] in addition to a well-established alternative, PPh2. Despite their differences, we found significant overlap in their predictions (p ≈ 2.46e−64 to 1.46e−53), supporting the compatibility of GeneEMBED with multiple impact estimators. The GeneEMBED framework crucially relies on the *PS* metric, which is compatible only with estimators that have probabilistic interpretations. While some tools (REVEL, SIFT, MutPred2, or VEST) fit this criterion, many do not have such interpretations or may require further transformations (e.g., CADD or Eigen).

The flexibility of the GeneEMBED strategy also applies to different networks. We found that GeneEMBED consistently identified similar genes across the three PPI networks used in this study (p ≈ 1.06e−8 to 5.78e−28; Figure S2), suggesting that usage of any well-constructed and disease-relevant network will tend to converge on similar findings. While the use of networks is key in the GeneEMBED strategy, it also introduces a potential source of error. Even stringently curated networks may be prone to research bias. Unbiased networks built with high-throughput techniques may provide alternatives. However, they tend to be limited in size due to technical constraints, resulting in an insufficient capture of disease-relevant interactions (STAR Methods; Tables S14, S15, and S16). In this regard, the GeneEMBED strategy showed robustness to the presence of both false-positive and false-negative edges (STAR Methods; Figure S7).

The flexibility of the framework also provides a channel for improvement in predictive power, namely, the edge-weighting scheme. While other edge-weighting approaches were characterized (STAR Methods; Tables S13, S14, and S15), the current framework estimates the perturbation of each interaction independently but considers all edges equally important. However, biological networks are highly robust to mutations due to pathway redundancies.[123,124] Among these, some are dominant while others are auxiliary,[125] suggesting that different parts of the network have varying levels of importance. This indicates a potential limitation and area for improvement in the GeneEMBED framework. Potential approaches to address this are to consider alternative methods of node embedding, including anisotropic diffusion techniques, which will be the focus of future work.

## Limitations of the study

While we anticipate GeneEMBED to be broadly applicable to case-control germline studies across a wide variety of complex genetic diseases, we note that this study is not without limitations. First, though we strove to validate the involvement of candidate genes in AD biology, further in-depth experimental characterization is necessary to elucidate their roles in pathology. Second, while the integration of network data is a key innovative component of GeneEMBED, it also presents a limitation. The reliance on network data means that, in the absence of interactions or genes that may be central to disease pathology, GeneEMBED may not make informative predictions. This suggests that GeneEMBED may have limited compatibility with certain networks (e.g., unbiased networks built on high-throughput screens). Third, the current implementation of GeneEMBED considers only coding mutations. However, a

growing body of literature suggests that non-coding changes may also play an important role in AD pathology,[126],[127] as well as other complex diseases. Therefore, extending GeneEMBED to incorporate non-coding data may be a fruitful future direction.

### Conclusions

In summary, using AD as a proof of concept, we show that, by placing genes in the context of their network interactions, GeneEMBED identifies novel disease genes that add to our understanding of pathology and which may harbor potential therapeutic value. This approach is general and can be applied to other sequenced case-control cohorts of a few thousands of subjects to decode gene variant interactions of interest in other complex genetic diseases.

### STAR★METHODS

Detailed methods are provided in the online version of this paper and include the following:

- KEY RESOURCES TABLE
- RESOURCE AVAILABILITY
  - Lead contact
  - Materials availability
  - Data and code availability
- EXPERIMENTAL MODEL AND SUBJECT DETAILS
  - Drosophila strains and neuronal dysfunction assay
- METHOD DETAILS
  - Whole exome/genome sequencing data
  - Variant scoring methods
  - GeneEMBED
  - Use of PCA vs full embedding distances
  - Downsampling analyses
  - Negative control experiment
  - Characterizing alternative weighting schemes
  - Sensitivity to false negative/positive edges
  - GeneEMBED performance in an unbiased network
  - Shuffled label experiments
  - MAGMA analyses
  - Recall of known AD genes
  - Network analyses
  - RNA sequencing analysis
  - Pathway enrichment analysis
  - Mouse phenotype analysis
  - Drug interaction analysis
- QUANTIFICATION AND STATISTICAL ANALYSIS

### SUPPLEMENTAL INFORMATION

### ACKNOWLEDGMENTS

We gratefully acknowledge support from the National Institutes of Health AG061105, AG068214, GM066099 (O.L.), AG074009 (O.L. and I.A.-R), and AG057339 (J.B. and J.M.S.). In addition, we acknowledge The Alzheimer's Disease Sequencing Project (ADSP), which is composed of two AD genetics consortia and three National Human Genome Research Institute (NHGRI)-funded Large Scale Sequencing and Analysis Centers (LSACs). The two AD genetics consortia are the Alzheimer's Disease Genetics Consortium (ADGC) funded by NIA (U01 AG032984) and the Cohorts for Heart and Aging Research in Genomic Epidemiology (CHARGE) funded by NIA (R01 AG033193); the National Heart, Lung, and Blood Institute (NHLBI); other National Institutes of Health (NIH) institutes; and other foreign governmental and non-governmental organizations. The Discovery Phase analysis of sequence data is supported through UF1AG047133 (to Drs. Schellenberg, Farrer, Pericak-Vance, Mayeux, and Haines); U01AG049505 to Dr. Seshadri; U01AG049506 to Dr. Boerwinkle; U01AG049507 to Dr. Wijsman; and U01AG049508 to Dr. Goate, and the Discovery Extension Phase analysis is supported through U01AG052411 to Dr. Goate, U01AG052410 to Dr. Pericak-Vance, and U01 AG052409 to Drs. Seshadri and Fornage. Data generation and harmonization in the follow-up phases are supported by U54AG052427 (to Drs. Schellenberg and Wang). The results published here are in whole or in part based on data obtained from the AD Knowledge Portal (https://adknowledgeportal.org). The Mayo RNA sequencing (RNA-seq) study data were led by Dr. Nilüfer Ertekin-Taner, Mayo Clinic, Jacksonville, Florida as part of the multi-PI U01 AG046139 (MPIs Golde, Ertekin-Taner, Younkin, and Price). Samples were provided from the following sources: The Mayo Clinic Brain Bank. Data collection was supported through funding from NIA grants P50 AG016574, R01 AG032990, U01 AG046139, R01 AG018023, U01 AG006576, U01 AG006786, R01 AG025711, R01 AG017216, and R01 AG003949; NINDS grant R01 NS080820; CurePSP Foundation; and support from Mayo Foundation. Study data include samples collected through the Sun Health Research Institute Brain and Body Donation Program of Sun City, Arizona. The Brain and Body Donation Program is supported by the National Institute of Neurological Disorders and Stroke (U24 NS072026 National Brain and Tissue Resource for Parkinson's Disease and Related Disorders), the National Institute on Aging (P30 AG19610 Arizona Alzheimer's Disease Core Center), the Arizona Department of Health Services (contract 211002, Arizona Alzheimer's Research Center), the Arizona Biomedical Research Commission (contracts 4001, 0011, 05–901, and 1001 to the Arizona Parkinson's Disease Consortium), and the Michael J. Fox Foundation for Parkinson's Research. AD Knowledge portal data were also generated from *post mortem* brain tissue collected through the Mount Sinai VA Medical Center Brain Bank and were provided by Dr. Eric Schadt from Mount Sinai School of Medicine. Further AD knowledge portal study data were provided by the Rush Alzheimer's Disease Center, Rush University Medical Center, Chicago. Data collection was supported through funding from NIA grants P30AG10161 (ROS), R01AG15819 (ROSMAP; genomics and RNA-seq), R01AG17917 (MAP), R01AG30146, R01AG36042 (5hC methylation, ATAC-seq), RC2AG036547 (H3K9Ac), R01AG36836 (RNA-seq), R01AG48015 (monocyte RNA-seq), RF1AG57473 (single nucleus RNA-seq), U01AG32984 (genomic and whole-exome sequencing), U01AG46152 (ROSMAP AMP-AD, targeted proteomics), U01AG46161(TMT proteomics), U01AG61356 (whole-genome sequencing, targeted proteomics, ROSMAP AMP-AD), the Illinois Department of Public Health (ROSMAP), and the Translational Genomics Research Institute (genomic). Additional phenotypic data can be requested at www.radc.rush.edu.

### AUTHOR CONTRIBUTIONS

Y.L., T.B., I.A.-R., K.L., J.B., and O.L. designed research; Y.L., T.B., I.A.-R., S.M., S.S., K.L., J.B., and O.L. performed research; I.A.-R., C.G.M., J.M.S., and J.B. contributed reagents and tools; and Y.L., T.B., I.A.-R., C.G.M., J.M.S., J.B., K.L., and O.L. wrote the paper.

### DECLARATION OF INTERESTS

The authors declare no competing interests.

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

**Cell Genomics**
Article

# STAR★METHODS

## KEY RESOURCES TABLE

| REAGENT or RESOURCE | SOURCE | IDENTIFIER |
|---|---|---|
| **Deposited data** | | |
| ADSP - Discovery Cohort Whole Exome Sequencing data | Alzheimer's Disease Sequencing Project ([128]) | phs000572.v8.p4; https://www.ncbi.nlm.nih.gov/gap/ |
| ADSP – Extension Cohort Whole Genome Sequencing data | Alzheimer's Disease Sequencing Project ([129]) | https://www.niagads.org/adsp/content/home |
| Post-mortem differential expression data for Alzheimer's disease | Accelerating Medicines Partnership – Alzheimer's Disease ([130–135]) | https://doi.org/10.7303/syn9702085 |
| Mouse Genome Informatics (MGI) database | Bult et al. ([88]) | http://www.informatics.jax.org/ |
| Drug-Gene Interaction database (DGIdb) | Freshour et al. ([136]) | https://www.dgidb.org |
| STRING protein-protein interaction network | Szklarczyk et al. ([44]) | https://string-db.org/ |
| HINT protein-protein interaction network | Das et al. ([45]) | http://hint.yulab.org/ |
| Brain specific protein-protein interaction network | Green et al.([46]), Zitnik et al. ([47]) | http://snap.stanford.edu/ohmnet/ |
| **Experimental models: Organisms/strains** | | |
| *D. melanogaster: UAS-Tau* | Lasagna-Reeves et al. ([49]), Bloomington *Drosophila* Stock Center | BDSC strain#: 51363 |
| *D. melanogaster: UAS-Aos:β* | Chouhan et al. ([50]), Bloomington *Drosophila* Stock Center | BDSC strain#: 33769 |
| **Software and algorithms** | | |
| Python 3.7 | Python Software Foundation ([137]) | https://www.python.org/ |
| GraphWave | Donnat et al. ([51]) | https://github.com/snap-stanford/graphwave |
| PolyPhen 2 variant impact scores | Adzhubei et al. ([43]) | http://genetics.bwh.harvard.edu/pph2/dokuwiki/start |
| Evolutionary Action variant impact scores | Katsonis et al. ([42]) | http://lichtargelab.org/software/dashboard/ |
| BCFTOOLS | Danecek et al. ([138]) | https://samtools.github.io/bcftools/bcftools.html |
| Eigenstrat | Patterson et al. ([139]) | https://reich.hms.harvard.edu/software |
| MAGMA | De Leeuw et al. ([68]) | https://ctg.cncr.nl/software/magma |
| GeneEMBED | This study | https://github.com/LichtargeLab/GeneEMBED https://doi.org/10.5281/zenodo.6654182 |

## RESOURCE AVAILABILITY

### Lead contact
Further information and requests for resources and reagents should be directed to and will be fulfilled by the lead contact, Olivier Lichtarge (lichtarg@bcm.edu).

### Materials availability
This study did not generate new unique reagents.

### Data and code availability
- This paper analyzes existing, publicly available data. These accession numbers for the datasets are listed in the key resources table.
- The code generated during this study is available at https://github.com/LichtargeLab/GeneEMBED (https://doi.org/10.5281/zenodo.6654182).
- Any additional information required to reanalyze the data reported in this paper is available from the lead contact upon request.

## EXPERIMENTAL MODEL AND SUBJECT DETAILS

### Drosophila strains and neuronal dysfunction assay
#### Genetics and strains

*Drosophila* lines carrying UAS-Tau, and UAS−Aos:β42 have been previously described[49,50] and are available from the Bloomington *Drosophila* Stock Center (BDSC, University of Indiana). For post mitotic pan-neuronal expression we used the elav-GAL4(C155) driver from BDSC. The alleles tested as potential modifiers targeting the *Drosophila* homologs of GeneEMBED candidate genes were obtained from the BDSC. Homologs were identified using BLAST and also the DRSC Integrative Ortholog Prediction Tool (Diopt score)[140,141] (Table S12). For the neuronal dysfunction tests, we used a highly automated behavioral (motor performance) assay based on the *Drosophila* startle-induced negative geotaxis response as previously described.[141,142] To assess motor performance of fruit flies as a function of age, we used 10 age-matched virgin females per replica per genotype. Four replicates were used per genotype. Flies are collected in a 24-hour period and transferred into a new vial containing 300μL of semi defined media (20g yeast, 20g Tryptone, 30g sucrose, 60g Glucose, 0.5g MgSO4˜7H2O, 0.5g CaCl2˜2H2O, 80g Inactive Yeast, 1L H2O) every day. Using an automated platform that uses a mechanized arm and clamp (https://nri.texaschildrens.org/core-facilities/high-throughput-behavioral-screening-core), the animals are tapped to the bottom of a plastic vials to trigger their negative geotactic response (climbing response) and are recorded for 7.5 seconds as they climb on the walls of transparent plastic vials. Videos are analyzed using custom software (code available for download on ref. [141]) that assigns movement trajectories to each individual animal, assesses their speed (mm/s) and returns an average per replicate per trial. Three trials per replicate are performed each day shown, and four replicates per genotype are used. A mixed effect model analysis of variance using spline regressions was run on Rstudio, using each four replicates to establish statistical significance across genotypes.[142] Human genes *POLD1* and *ANLN* were identified as modifiers in a separate manuscript currently under revision and were not directly tested here. All shown modifier alleles had a significant effect (p <0.01) compared to the disease controls.

## METHOD DETAILS

### Whole exome/genome sequencing data

Whole exome sequencing (WES) data from 5,169 individuals were downloaded from NIH NCBI study ID: phs000572.v8.p459 (ADSP Discovery)[128] and a further 969 whole genome sequences (WGS) were downloaded from National Institute on Aging Genetics of Alzheimer's Disease Data Storage Site (NIAGADS) dataset: NG0006760 (ADSP Extension).[129] Samples making up the Discovery and Extension cohorts were selected from a set of 24 well characterized cohorts from the Alzheimer's Disease Genetics Consortium. The sample phenotypes were coded as 0 or 1 indicating non-AD and AD, respectively in both cohorts. Only samples of European/White ancestry were used in the analyses of both cohorts. The mean age of AD onset for AD positive Discovery cohort samples was 75.3 years with a standard deviation of 8.3 years, while the mean age of last exam for control samples was 85.5 years and a standard deviation of 5.1 years. The mean age of AD onset for AD positive Extension cohort samples was 75.5 years with a standard deviation of 7.8 years. Healthy controls of the Extension cohort had a mean age at last exam of 75 years with a standard deviation of 8.3 years.

### Quality control and annotation of WES and WGS data

Although extensive QC procedures were performed on the WES Discovery and WGS extension cohorts by the ADSP and GCAD consortia,[143] respectively, we generated QC statistics for Ti/Tv, number of variants, singletons, and missingness for each sample and HWE, genotyping rate (AC/AN) for each variant site across cases and controls. Then, potentially false-positive variants sites and outlier samples were removed. HWE (Hardy Weinberg Equilibrium) exact test[144] was performed on the control samples of each cohort and the variants with HWE violations (HWE p-value < 5E-8) were removed. We also removed the variants that were genotyping rate less than 0.95 in either case and control and in combined case and control samples. Outlier samples including potentially non-whites were identified based on Ti/Tv, total number of variants, singletons, and missingness. To cluster samples with genetic background and identify outliers of clusters, we applied Principal Component Analysis (PCA) method. We identified potentially related samples by estimating genetic relationships between samples with kinship coefficients. We removed outliers that include non-European descendants. To annotate consequences of variants, we used the Annovar133. Then, non-synonymous single nucleotide variants (SNVs) and small indels, which lead to frames-shift, excluding CNVs (copy number variants) were annotated with EA. BCFTOOLS,[138] KING,[145] and SMARTPCA from Eigenstrat package were used for calculating variant and sample statistics, inferring relationships, and for estimating sample clusters with PCA,[139] respectively.

### Variant scoring methods

Two variant scoring methods were used to describe the mutational impact of variant, separately. The first of these two methods are PolyPhen2 (PPh2), which predicts the potential impact of an amino-acid substitution on protein function using a machine learning algorithm trained on sequence and structural information. Here, PPh2 HDIV raw scores were used. PPh2 scores range from 0–1, where increasing value indicates increasing severity of mutation.

The second scoring method we used was Evolutionary Action (EA), which expresses that the genotype-phenotype relationship can be written as $f(\gamma) = \varphi$, where evolutionary fitness function (*f*) maps genotype (γ) onto fitness landscape (φ). SNVs are considered

small perturbation in the genome ($d\gamma$) and cause perturbation in fitness ($d\varphi$) : $\nabla f \cdot d\gamma = d\varphi$. A missense mutation at residue $r_j$, $d\gamma \approx \Delta r_j$, will cause all components of $\nabla f$ to be forced to zero except $\frac{\partial f}{\partial r_j}$, and impact equation simplifies to $\Delta\varphi \approx \frac{\partial f}{\partial r_j}\Delta\gamma$. Evolutionary Trace[146] is used to compute $\frac{\partial f}{\partial r_j}$, and $\Delta\gamma$ can be approximated with amino acid substitution log-odds ratios. EA scores are reported between 0-1 with increasing severity of functional impact, where EA = 0 indicates no effect on protein function and EA = 1 indicates loss of function. In the EA scoring systems, silent mutations are given a score of EA = 0, while frameshift and stop mutations are given a score of EA = 1.

### GeneEMBED
#### Network construction
In the bulk of the work presented here, we use three biological networks for protein-protein interactions including STRING v10 [44], HINT,[45] and a brain specific network.[46,47] The STRING network defines edges between genes using many forms of evidence including curated interactions, experimental interactions, protein homology, co-expression, text mining, etc. The HINT network consists of manually and systematically curated edges requiring interactions to have been reported at least twice in literature. The brain specific network consists of genes who demonstrate tissue specificity per Human Protein Reference database and BRENDA Tissue ontology. Edges in the brain specific network are listed only if there is experimental evidence for an interaction. For in-depth construction details, please see the appropriate publications. For use in this approach, all edge confidence scores in all networks were removed and replaced with a weight of 1, simply indicating an edge exists between two genes.

Networks are first made sample specific by integrating mutational information. First we compile the functional effect of a set of variants in an individual into one gene level score called a perturbation score (*PS*), defined as: $PS_{gene} = 1 - \prod_{i}^{v}(1 - VIS)^{zyg}$, where

$v$ is the number of variants in a gene for the individual, $i$ is the index over those variants and $zyg \in \{0, 1, 2\}$ where 0 denotes wild type, 1 denotes heterozygous, and 2 denotes homozygous for variant $i$, and *VIS* denotes functional impact of variant (EA or PPh2 score). To construct sample specific networks, we calculate edge weights as the sum of the *PS* of the two connected genes: $W_{edge} = |PS_x + PS_y|$. Characterization of alternative edge weighting schemes and their corresponding discussions can be found below and Tables S17, S18 and S19. Finally, to construct disease and control specific networks, edge weights are averaged over all cases and controls separately to build a case specific and control specific mutation weighted network.

#### Node embedding algorithms
In order to assess network perturbations in genes between cases and controls, we use the GraphWave algorithm[51] to generate node embeddings. The GraphWave algorithm has advantages over other embedding algorithms in that it provides rigorous mathematical guarantees on identifying structure preserving embeddings. GraphWave performs unsupervised node embedding on node structure (i.e. topological patterns of node connectivity). Accordingly, the authors provide proof for the equivalency of embeddings between two structurally identical nodes a and b, which rests on the assumption that there exists a one-to-one mapping between the K-hop neighborhood of the two nodes. We can extend this proof to claim that the embeddings of a node from two identical graphs must also be equivalent since there will exist a mapping between the node neighborhoods. Thus, when comparing disease and healthy graphs wherein the node connectivities are largely unchanged, the descriptive features captured by each dimension in the embedding space are the same, thus allowing for direct comparisons. The GraphWave algorithm is briefly described below.

Let $V$ denote the eigenvectors and $\lambda_n$ denote the eigenvalues ($\Lambda = diag(\lambda_1, \lambda_2, ..., \lambda_n)$) of the graph Laplacian $L = D - A = V\Lambda V^T$, where $D$ denotes the degree matrix and $A$ denotes the adjacency matrix of the graph. Now consider a low-pass filter kernel $g_s = e^{-\lambda s}$, where $s$ is some scaling factor, we may define spectral graph wavelets by modulating the graph Laplacian by kernel $g_s$:

$$\Psi_a = V\Lambda(g_s(\lambda_1), ..., g_s(\lambda_n))V^T\delta_a$$

where $\delta_a$ is a Dirac signal about node $a$, $\Psi_a$ is an n-dimensional vector representation of the spectral graph wavelet of node $a$, and $s$ is a scaling factor corresponding to the radius of the neighborhood around node $a$. GraphWave samples over a set of $s_j$ for $s_j \in \{s_{min}, s_{max}\}$ where $s_{min}$ and $s_{max}$ are automatically calculated.

We can recover coefficients of the graph spectral wavelet $\Psi_a$ corresponding to a neighbor node $m$ by:

$$\Psi_{ma} = \sum_{i=1}^{N} g_s(\lambda_i)V_{mi}V_{ai}$$

Where $\Psi_{ma}$ represents the signal received by node $a$ from a neighbor node $m$, and $V_{mi}$, $V_{ai}$ denote the $i$-*th* value of the eigenvectors of $m$ and $a$. Similarities in node characteristics are carried in $\Psi_{ma}$ coefficients. GraphWave proposes to use the $\Psi_{ma}$ coefficients as components of a characteristic function, which, when sampled at $d$ evenly spaced points, allows a 2-d representation of $\Psi_a$:

$$\varphi_a(t_j) = \frac{1}{N}\sum_{m=1}^{N} e^{it\Psi_{ma}}$$

where $t_j$ comes from the set of $d$ evenly spaced points $(\{t_1,t_2,\ldots,t_d\})$, and $i$ is the imaginary unit $(i = \sqrt{-1})$. The final embedding of the node is then collected as:

$$X_a = [Re(\varphi_a(t_j)), \; Im(\varphi_a(t_j))]_{t_1\ldots t_d}$$

### Gene identification

In order to find genes with differentially perturbed network characteristics between case and control, GraphWave is applied to both networks. Each gene will now have two embeddings, one corresponding to the case network and another from the control network. The GeneEMBED hypothesis supposes that genes contributing to disease will have significantly differing case and control embeddings. To prioritize genes accordingly, we perform principal component analysis (PCA) on the node embeddings and measure distances between case and control embeddings in the PCA space. The role of PCA in the methodology is to aid in denoising the full dimensional embeddings retrieved by GraphWave. The full dimensional embeddings produced by the algorithm can encompass signals ranging from immediate neighborhoods to the complete graph. As a result, the full dimensional embedding of a node will be influenced by any change in the edge weight anywhere in the graph. In order to remove some of these noisy influences, we perform a PCA on the full dimensional embeddings and use the first principal component to measure distances as this component recovers between 78-92% of the variability explained. Characterization of performance between distances computed on full dimensional embedding against distances measured on PCA is shown in Tables S17, S18 and S19 and discussed further below. By defining distances as the square root of the L2-norm of each gene measured between case and control, we are able to reconstruct a gaussian-like distribution from the positive and negative values. We then compute z-scores and their corresponding one-tailed Z-test p-values for each distance value relative to the full distribution. Then, we perform false discovery rate (FDR) corrections on the p-values using the Benjamini-Hochberg method and genes corresponding to distance values passing FDR <0.01 are selected as pre-candidate genes. Lastly, the full GeneEMBED process is performed on healthy controls vs healthy controls (details of healthy control selection are discussed below). Genes passing FDR <0.01 threshold in this control vs control analysis are removed from the list of pre-candidate genes. This is done to filter potential sources of variation which may not be disease specific (false positives, discussed below, Tables S20 and S21). The final set of genes passing FDR <0.01 threshold and not removed by control vs control analysis are considered candidate genes.

### Computational efficiency/requirements

GeneEMBED offers an analytical framework to appraise all coding genes in the human genome with respect to their attributes in a molecular network. Accordingly, this can be computationally demanding depending on the size of the network being used. In this study we used three different PPI networks, a brain specific network, HINT, and STRING. After annotation and preprocessing of exomic variant calling file (VCF), the computational time required for the brain network consisting of 3.2k nodes and 48k edges was 649 seconds (10.8 minutes). Similarly, for the HINT network consisting of 12.6k nodes and 146k edges, the computational time from annotation of networks with mutational information to identification of candidate genes was 3058 seconds (51 minutes). Lastly, for the STRING network consisting of 15k nodes and 1.9m edges, the computational time was 6.2 hours. All network analyses were performed on a server with specifications of Intel Xeon Gold 5222 CPU at 3.8GHz with 8 cores and 348gb RAM. GeneEMBED is implemented in python 3[137] and is publicly available as noted in the Data and Code Availability section.

### Use of PCA vs full embedding distances

In order to assess the effects of PCA on the GeneEMBED methodology, we tested the utility of computing distances based on the full dimensional embedding outputs from the GraphWave algorithm compared to PCA-distances. We ran GeneEMBED with both weighting metrics on the Discovery-VISEA cohort using the STRING network and recovered 82 genes with full embedding distances and 69 with PCA-distances. To test the relevance of the identified gene to AD, we measured their: (i) recovery of AD-associated genes, (ii) connectedness to known AD-associated genes, and (iii) differential expression in postmortem AD brains. We found that distances based on full dimensional embeddings were able to recover statistically significant overlaps (one-tailed hypergeometric test) with GWAS Meta 1, GWAS Meta 2, and DisGeNet. PCA-distance gene set recovered significant overlaps with DisGeNet and CTD (Table S13). Next, we found that genes identified by full dimensional embeddings were significantly connected to GWAS Meta 1, GWAS Meta 3, and CTD gene sets. Comparatively, PCA-distance gene set showed significant connectivity to all five reference gene sets (Table S14). Finally, genes identified by full dimensional embeddings showed no statistically significant enrichment for differential expression in post-mortem AD brains, while PCA-distance gene set was enriched in two brain regions with significance (permutation-based Z-test p = 0.012) (Table S15). These data show that distances based on PCA performs better than full dimensional embeddings. Further examination of the genes identified by both approaches showed that 74% of genes identified by the full dimensional embeddings were also identified in the PCA-distance framework. However, of the 20 genes unique to the full dimensional embeddings' gene set, only 3 were dysregulated in AD brains. Comparatively, of the 7 genes unique to the PCA-distance framework, 4 were dysregulated in AD brains. Overall, this demonstrates the role of principal component analysis in denoising the raw outputs of the GraphWave algorithm.

### Downsampling analyses

While large sequencing cohorts are becoming more commonplace in recent years, for some rarer phenotypes and diseases, it is still a challenge to produce such sample sizes. In order to characterize its performance at various cohort sizes, we performed an iterative

downsampling analysis of GeneEMBED (detailed in methods) on the Discovery cohort. Using the gene set identified with the full cohort as ground truth, we calculated precision and recall of GeneEMBED gene sets identified at sub-cohort sizes of 80%, 60%, 40%, 20%, 10%, and 5% of the original cohort. Performing this experiment with both EA and PPh2, we found that sub-cohort size could be dropped as far as 40% of the original cohort before recall fell below 0.6 for PPh2 and EA analyses (Figure S6A).

Next, we examined the relationship between network connectivity and gene identification by GeneEMBED at various cohort sizes. We correlated the PCA-distances based ranking of identified genes with their ranking by betweenness, degree, and eigen centrality using STRING network. We measured correlation with the Spearman rank-order correlation test. We found that for both EA and PPh2 based analyses, the correlation with these centrality measures was relatively stable regardless of the decrease in cohort size (Figure S6C). Together, these data suggest that the utility of GeneEMBED is not limited to large cohort sizes but can sufficiently extended to much smaller cohort sizes.

### Negative control experiment

In order to characterize the behavior of GeneEMBED in the absence of disease specific mutational information, we performed the analysis on only healthy controls from the Discovery cohort. Healthy samples were defined to be individuals, from the ADSP Discovery cohort, who were homozygous for the APOE$\varepsilon$3 variant, and had low BRAAK staging (1 or 2). This filtering resulted in 725 control samples which were randomly split into two groups. We then tested the Spearman rank-order correlation of genes which pass FDR threshold with ranks generated by connectivity measures used above. Strikingly, we observed that in GeneEMBED analysis using the STRING network, the PCA-distances correlated with degree, betweenness, and eigenvector centralities at more than twice the rate in healthy vs healthy (r = 0.273, 0.215, 0.282) than in case vs control analyses (r = 0.121, 0.085, 0.103), and that correlations for healthy vs healthy were all statistically significant while the case vs control analyses were not (Figure S6B). Even more notable were the disparities of correlations in the Brain Specific PPI network, correlations with degree, betweenness, and eigenvector centralities in the healthy vs healthy analysis (r = 0.596, 0.481, 0.630) which were all statistically significant and case vs control analyses (r = 0.111,0.358,-0.103) which were not significant. These findings were again echoed in the HINT network analyses where healthy vs healthy gene set correlated significantly with network centrality measures. The findings suggest that in the absence of disease-relevant mutational data, GeneEMBED prioritizes genes with large network connectivity as small mutational differences are likely amplified by the gene's network influence.

### Characterizing alternative weighting schemes

In order to characterize the performance gain or drop-off of GeneEMBED to differences in edge weighting schematics, we tested two alternative weighting approaches. The first alternative approach was to assign the edge weight between two nodes as the maximum PS score between them (max(PS)). The second approach was to assign edge weight between two nodes based on a PS threshold. If either one of the connected genes had a PS above PS > 0.7 threshold, then the edge is considered dead, otherwise the weight is determined by the previous method. We ran GeneEMBED with both weighting metrics on the Discovery-VIS$^{EA}$ cohort using the STRING network and compared outputs with the original framework which uses sum(PS). We recovered 73 genes from max(PS) and 72 genes from PS threshold. Interestingly, nearly 90% of identified genes overlapped with the sum(PS) based gene set. To test the relevance of the two gene sets to AD, we measured their: (i) recovery of AD-associated genes, (ii) connectedness to known AD-associated genes, and (iii) differential expression in postmortem AD brains. Both max(PS) and PS threshold schemes recovered statistically significant overlaps in the DisGeNet reference gene set (Table S13). Comparatively, the current weighting scheme, sum(PS), recovered similar statistically significant overlaps in DisGeNet as well as significant overlaps with the CTD gene set. Next, max(PS) showed significant diffusion to ClinVar and GWAS Meta 1 gene sets, while PS threshold showed significant diffusion to all gene sets except CTD (permutation-based Z-test, Table S14). Comparatively, sum(PS) showed statistically significant diffusion to all gene sets. Lastly, we found that genes identified by max(PS) were enriched for differential expression in one brain region out of seven, though this was not significant (permutation-based Z-test p = 0.076). The genes identified by the PS threshold scheme, like sum(PS), were enriched for differentially expressed genes in two brain regions (permutation-based Z-test p = 0.0014 and 0.012 respectively) (Table S15). These data suggest that though the three weighting schemes overlap in close to 90% of their identified genes, sum(PS) performs better than alternative weighting methods. This is because sum(PS) allows each interaction to be described independently. For example, consider a protein with many interactions. If the protein has a mutation in some binding site, this may lead to a high PS due to the inactivation of the binding site. In this case, the interaction between the protein and any interactor which is mediated by the mutated binding site is significantly perturbed. However, this does not necessarily mean that the protein's interactions mediated by other unaffected binding sites are perturbed to the same degree. Sum(PS) allows for this flexibility while the other weighting schemes do not, which leads to its improved performance.

### Sensitivity to false negative/positive edges

While the curation of biological networks has become increasingly more sophisticated, it is important to recognize that even networks built upon stringent curation of experimentally validated edges may be prone to research bias. In order to characterize the sensitivity of GeneEMBED to false negative/positive edges, we applied GeneEMBED to the Brain Specific network while iteratively deleting (or adding) edges. Candidate genes identified by GeneEMBED on the unmodified Brain network using the Discovery-VIS$^{EA}$ cohort were used as ground truth for comparison.

Sensitivity of GeneEMBED to false negative edges was assessed by iteratively and randomly deleting edges in the brain network in increments of 5% from 5% to 70%. At each increment, GeneEMBED was applied to the modified Brain network and identified candidate genes were used to measure recall and precision relative to ground truth as defined above. We found that up to 55% of the edges in the original network could be randomly deleted before either recall or precision fell below 0.6 (Figure S7A). Moreover, when we restricted random deletion of edges to those involving any of the genes identified in the unaltered Brain specific PPI network, we found that up to 30% of their edges could be deleted before either recall or precision fell below 0.6 (Figure S7A).

Sensitivity of GeneEMBED to false positive edges was assessed by iteratively and randomly adding edges to the Brain network in increments of 5% from 5% to 110%. Percentage is measured relative to the size of the unmodified Brain network and the pool of possible edges to add was taken from the full set of edges required to make the network fully connected. Recall and precision of candidate genes identified on the modified network are measured in the same manner as specified above. We found that we could randomly add edges totaling up to 80% of the original network size (~38.4k edges) before either the precision or recall fell below 0.6 (Figure S7B).

These data suggest GeneEMBED is highly robust to both false positive and false negative edges. In the case of random deletion of edges (FN edges), it is likely that there are more genes that do not play a role in AD pathobiology than genes that contribute significantly to pathogenesis. Accordingly, there will be more edges that are not associated with AD than edges that are associated with AD. Therefore, it is possible to randomly delete a large number of edges while maintaining a high recall and precision. However, when there is a bias in the edge deletion process to informative edges, the methodology becomes more sensitive to FN edges. Similar reasoning can be applied to the case of random edge addition (FP edges), as there are likely more edges that are not associated with AD it is possible to have large numbers of FP edges before recall or precision drop below 0.6. Overall, these data show that while there may be potential research bias in curated biological networks, the strategy employed by GeneEMBED allows for its robustness to the presence of false positive and false negative edges.

## GeneEMBED performance in an unbiased network

In order to benchmark the GeneEMBED strategy with a network without any functional bias or literature curation, we employed the HuRI network.[130] The HuRI network is the largest unbiased interactome map of binary protein-protein interactions. The network contains 8,275 nodes and 52,569 edges generated from an impressive array of nine different 'all-by-all' screens of 17,408 proteins. Using this network as a starting point, we ran GeneEMBED using the Discovery-VIS$^{EA}$ cohort and identified a candidate gene set. To test the relevance of the identified genes to AD biology, we examined: (i) direct overlaps with reference gene sets discussed previously, (ii) connectedness between reference gene sets and identified genes, and (iii) dysregulation of identified genes in postmortem AD and non-AD samples from the AMP-AD dataset. Performing these experiments, we found that there was no significant recovery of known AD-associated genes. We also found no significant preferential connectivity between candidate genes and known AD-associated genes (Tables S16 and S17). We did find an enrichment of the candidate genes for differentially expressed genes in AD vs non-AD brains with marginal significance (permutation-based Z-test $p = 0.06$) (Table S18). While these results would seem to suggest that GeneEMBED is unable to perform on such unbiased networks, it is important to consider the HuRI network in the context of AD. Despite being the largest of its kind to date, due to technological limitations, the HuRI network comprises only half the exome. Accordingly, only half or less of the genes in the reference gene sets were present in the HuRI network. Indeed, several genes which are core to AD pathobiology, such as APOE, TREM2, or MAPT, were absent in HuRI. The stringency of the HuRI network's construction suggests that while it has a low FP rate, it may be depleted in protein-protein interactions. Indeed, we have observed that GeneEMBED is more robust to FP edges than FN edges (Figure S7). Overall, these data emphasize the importance of appropriately selecting a starting network. While it is recommended to use an unbiased network when possible, it is also crucial to ensure the network is reflective of the biology of the target disease.

## Shuffled label experiments

In the presence of counterproductive mutational data and large influence from network inputs, similar genes will be recovered from various shuffled label experiments leading to inflated overlaps. Indeed, it is likely that due to ambiguous mutational input the identified overlapping genes from randomly shuffled trials are less related to AD than case vs control overlaps. Moreover, the large reliance on network information suggests that identified gene lists may be strongly correlated with network connectivity. To test this hypothesis, for VIS$^{EA}$ and VIS$^{PPh2}$ separately, we shuffled the labels (case or control) of individuals in both cohorts and applied GeneEMBED, repeating this 5 times due to computational time of the approach. We then measured, the number of genes overlapping between GeneEMBED applied to shuffled Discovery cohort and shuffled Extension cohort, giving 25 pairwise comparisons.

We found that for VIS$^{EA}$, an average of 31.7 genes overlapped among gene sets identified using label shuffling for the Discovery and Extension cohorts. Comparatively, an overlap of 14 genes was observed in the original framework after removing potential FPs from control vs control analysis. Importantly, we found that the 14 genes identified in the original analysis showed significant hypergeometric overlap with all five of the reference gene sets of known AD-associated genes ($p = 0.019–0.0039$). The overlaps identified by shuffled labeling showed few to no significant overlaps with any of the five reference gene sets (Table S19). Next, to further assess the relationship of the identified overlaps to AD, we performed literature curation analyses. For each gene in a gene set, we queried the PubMed database for publications co-mentioning the genes with AD in abstracts. Genes were only considered related to AD if they had at least 5 co-mentions. Statistical significance was determined using a permutation testing strategy. We randomly

generated 50 gene sets of the same size and counted the number of genes that were related to AD. We then measured the z-score of the observation relative to the background. We found that the original observation of 14 overlapping genes had a z-score of 6.86. Comparatively, the overlaps identified by random shuffling had an average z-score of 3.43 and stdev of 1.57. Lastly, we found that ranked gene lists derived from random shuffling were significantly correlated with degree centrality (Pearson correlation coeff. = 0.2–0.36, p = 0.037–1.7e-5), whereas the gene list derived from case vs control analysis was not correlated with Pearson correlation coefficient of 0.085 and a pvalue = 0.43.

Similarly, in VIS[PPh2] analysis, an average of 37.5 genes overlapped among gene sets identified by label shuffling. In contrast, 16 genes were found overlapping between Discovery and Extension cohorts using the original framework after removing potential FPs. While no significant overlaps were observed with the reference gene sets (Table S20), we found that the PubMed literature curation analysis showed significant association of overlap genes identified from case vs control analysis to AD with a z-score of 4.49. In comparison, overlaps obtained from label shuffling had an average z-score of 0.97 and stdev of 0.82. These data suggest that overlaps observed between Discovery and Extension cohorts in the original analysis are much smaller than expected by label shuffling trials. Despite these large differences in sizes, overlaps from the original framework are more related to AD. Further, they tend to rank genes independently of their pure connectivity, whereas label shuffling leads to a heavy dependence on network information. Overall, these observations demonstrate that during a lack of informative mutational data, GeneEMBED will tend to depend heavily on network information, identifying genes which are less relevant to AD than genes identified through productive (case vs control) mutational data.

### MAGMA analyses

We used MAGMA as a methodological control and analysis was performed on the same datasets. The variants were annotated with each corresponding NCBI reference genes of GRCh37 or 38. Next, we calculated each gene's one-sided regression p-values based on the snp-wise Mean model with a '–burden flag' to avoid deteriorating power of extreme rare alleles and the allele frequency threshold '0.1'. A threshold of p < 0.001 was used because the FDR thresholds resulted in too few genes for meaningful comparison to GeneEMBED.

### Recall of known AD genes

To test whether our approach could recover known genes related to AD we assessed direct overlaps. five gene sets were used to define known AD related genes: Comparative Toxicogenomic Database (CTD) gene set of 103 AD related genes,[71] a set of 25 genes identified by meta-analysis of large scale GWAS of diagnosed AD (GWAS Meta 1),[5] a set of 38 genes identified by another meta-analysis of AD GWAS studies (GWAS Meta 2),[6] a set of 208 genes with associations to AD from DisGeNET (DGN),[70] and a set of 21 genes acquired from the ClinVar database.[78] Significance of direct overlaps was assessed with one-tailed hypergeometric tests between sets of known AD genes and candidate gene sets.

### Network analyses

nDiffusion[82] was applied to measure how well GeneEMBED candidates were connected to known AD genes (defined above). nDiffusion relies on graph information diffusion methods[79–81] wherein signals are propagated from genes of interest to all genes in a network through their connections. Genes that receive more signal are more connected to genes of interest. Therefore, if known AD genes receive more diffusion signal from GeneEMBED candidates than other genes in the network, they are more connected. This diffusion connectivity is quantified by measuring the area under receiver operating characteristic curve (AUROC). nDiffusion also selects random sets of genes with similar degrees of connectivity as genes of interest and measures their AUROC. This permutation is performed 100 times to produce a background random distribution of AUROC, from which z-scores of the experimental observation are calculated using a one-sided Z-test. Two sets of genes are then deemed significantly connected if their AUROC is >0.5 and has a z-score > 2. The nDiffusion webtool was used to perform these analyses with the preset default settings.

### RNA sequencing analysis

In order to assess whether expression changes of GeneEMBED candidates in AD brain tissue, we used the AMPAD data sets.[74,131–135] Significant differentially expression (DE) was defined, per brain region, as genes which had log2(fold-change) > 0.263 or log2(fold-change) < 0.263 and FDR <0.05, as measured by AMPAD. This thresholding provided 1880 DE genes for cerebellum (CBE), 2952 genes for temporal cortex (TCX), 56 genes for frontal pole (FP), 73 genes for inferior frontal gyrus (IFG), 1579 genes for parahippocampal gyrus (PHG), 271 genes for superior temporal gyrus (STG), and 161 genes for dorsolateral prefrontal cortex (DLPRC). AD case versus non-AD control differential expression analysis results from all brain regions listed above are available online (https://doi.org/10.7303/syn9702085). To assess whether GeneEMBED candidates were enriched for DE genes, we performed one-tailed hypergeometric tests per brain region. These hypergeometric tests were limited only to the set of genes which were present in both the RNA-sequencing data from AMP-AD cohort and the WES data from the ADSP Discovery and Extension cohorts (i.e. only genes sequenced in both data sets were used). We performed these tests over all seven brain regions to identify region specific enrichments. Next, to determine statistical significance of having enrichment in n out of seven brain regions, we adopted a permutation testing strategy. We repeated the above analysis 1000x with randomly selected gene sets of similar size as candidate gene set. One-tailed Z-test p-values

were calculated for the observed number of enriched regions in candidate gene set relative to the distribution observed from random gene sets. This analysis was repeated for all GeneEMBED candidate gene sets, GWAS Meta 1 gene set, and GWAS Meta 2, gene set.

### Pathway enrichment analysis

Protein-protein interaction network of high confidence GeneEMBED candidates was built with the *Homo sapiens* STRING v11[147] using the combined score of all evidence types at a threshold of 0.400. *HiDef-Louvain* algorithm tool in the Community Detection extension algorithm of Cytoscape was used for clustering followed by functional enrichment analysis of each of the 21 main clusters. Gene set enrichment analysis was performed using the *iQuery*, *EnrichR* and *Gprofiler* community detection interfaces. Enrichments were considered significant if gene set FDR <0.05. Network was represented using Cytoscape v3.8.2 [152].

### Mouse phenotype analysis

To assess the relationship between high-confidence GeneEMBED genes and mouse phenotypes, we downloaded the files VOC_Mammalian_Phenotype.rpt and HMD_HumanPhenotype.rpt from the Mouse Genome Informatics (MGI) database (downloaded Nov. 2021). Within the downloadable database, we queried our full set of 143 genes and found that only 139 were documented in the database. These 139 genes mapped to 182 mouse homologs/orthologs. We then tallied the number of mouse genes in our candidate set which had annotations for the high-level mammalian phenotype of 'Nervous system phenotype'. We then tallied the total number of mouse genes in the downloadable database which had the same high level mammalian phenotype annotation. We then performed a one-tailed Fisher's Exact Test to determine the statistical significance of our observations. Additionally, we repeated this analysis for high level mammalian phenotype categories of (i) 'Behavioral/Neurological phenotype' and (ii) 'Nervous system phenotype' AND 'Behavioral/Neurological phenotype'.

### Drug interaction analysis

To assess whether any of our high confidence candidate genes were potential therapeutic targets, we used the Drug-Gene Interaction database (DGIdb).[148] The set of high-confidence candidate genes were input into the 'Search Drug-Gene interactions' webtool on the DGIdb website. We applied preset filters of 'Approved' indicating FDA-approved drugs only. We then filtered the subsequent list of drug-gene interactions for those which were annotated as having a directional (inhibiting or activating) effect. The resulting genes were then queried through PubMed database for co-mentions with 'Alzheimer' in abstracts.

### QUANTIFICATION AND STATISTICAL ANALYSIS

All of the quantitative and statistical methods, strategies, and analyses are described in the relevant sections of the method details or in the table and figure legends.

