## [Document S2. Transparent peer review records for Lagisetty et al. · Cell Genomics]

Identification of Risk Genes for Alzheimer's Disease by Gene Embedding

Yashwanth Lagisetty^{1,2}, Thomas Bourquard², Ismael Al-Ramahi^{2,3,4}, Carl Grant Mangleburg², Samantha Mota², Shirin Soleimani², Joshua M. Shulman^{2,3,4,5,6}, Juan Botas^{2,3,4}, Kwanghyuk Lee², Olivier Lichtarge^{2,4,7,8}*

Summary

Initial submission: Received : 2/17/2022

Scientific editor: Sonia Muliyil, Laura Zahn

First round of review: Number of reviewers: 3
Revision invited : 4/5/2022
Revision received : 5/18/2022

Second round of review: Number of reviewers: 3
Accepted : 7/3/2022

Data freely available: Yes

Code freely available: Yes

This transparent peer review record is not systematically proofread, type-set, or edited. Special characters, formatting, and equations may fail to render properly. Standard procedural text within the editor's letters has been deleted for the sake of brevity, but all official correspondence specific to the manuscript has been preserved.

Referees' reports, first round of review

Reviewers' Comments:

Reviewer #1:

CELL-GENOMICS-D-22-00033 "Identification of risk genes for Alzheimer's disease by gene embedding" by Lagisetti et al. In this m/s, the authors present an approach to identify Alzheimer's Disease (AD) related genes based on the comparison of the functional impact of genetic variants in biological networks derived from healthy and disease patient cohorts. Then they use the two networks to identify proteins whose interactions are significantly and exclusively perturbed in the AD cohort. They used GraphWave, a network embedding technique which learns low dimensional vectors for each protein based on the network topology and annotations. By comparing the protein embeddings between the healthy and disease cohorts they identify proteins that are significantly distinct between cohorts, suggesting a list of proteins potentially associated with AD. The authors show that this method produces statistically significant robust gene sets across different cohort sources and show multiple connections between the identified genes and the known AD biology, such as the overlap with known AD gene sets, dysregulation of these genes in relevant tissue and enrichment in AD associated biological processes. Finally, they validate some of the high-confidence genes in AD-Drosophila models, where they test the neuronal disfunction effect of modulating the expression level of the fly homologs to the genes identified by GeneEMBED. The m/s is well structured and clearly written and, while most of the methodology used in the approach is already published, the proposed strategy is novel and original. Furthermore, the authors showed it can be used on small sample sizes, an important limitation for the study of many diseases such as Alzheimer, providing results that were more robust than other state-of-the-art methods. Thus, overall, I am happy to support its publication. However, there are a few points that should be addressed and clarified, to improve the reproducibility of the work and increase its impact. In particular, I found difficult to follow the strategy used to embed the networks and derive the AD related genes. Additionally, even though the reported results are statistically significant, some of the conclusions seem slightly over optimistic regarding the accuracy of the method. Please, find below some specific comments that, I hope, might improve the clarity and robustness of the m/s.

As stated above, the methods section referring to the embedding pipeline requires some clarification, and it also misses important technical details that required for a complete understanding of the methodology. More specifically:

* Given that GraphWave embeddings are obtained separately for the disease and healthy cohorts, it is not evident if their dimensional spaces can be directly compared (i.e. whether each dimension in the embedding spaces capture equivalent features) since, as far as I know, GraphWave is not an inductive embedding method.

* The authors justify the use of the PCA with the following state: "This step is necessary because embeddings generated from GraphWave represent complex functions rather than a point in a multidimensional space". However, according to the GraphWave implementation provided by the original authors in their github (<https://github.com/snap-stanford/graphwave>), the algorithm already provides embeddings in the Euclidean space through the characteristic function. Thus, it is not clear which is the role of the PCA in the methodology as the given justification does not agree with the implementation reported by the GraphWave authors. It would be interesting to see if there are significant differences between distances calculated from the principal components and those extracted from full dimensional embeddings.

* Along the same line, the authors do not specify the number of PCA dimensions used to calculate the distances (i.e. the number of dimensions/features used to calculate distances between embeddings). Besides, the variability explained by the used dimensions should be also reported as it helps quantify the information that was eventually used from the learnt GraphWave embeddings.

* It is not clear how the authors derived a FDR value from the distance distribution. FDR correction is usually used to correct p-values but it seems it was directly implemented, somehow, to the PCA-distances. This step would also require further clarification.

STRING networks are always dangerous when it comes to interpretation since they are built from many

different types of data, including text mining and functional associations. Thus, validating results obtained using STRING networks through reported functional associations (i.e. AD-related genes) is tricky, since this information could have been included in the initial network. Similarly, the authors validate their embeddings strategy by comparing it to a network built from HINT, applying a heat-propagation algorithm (i.e. HotNet2) and computing the distances between nodes in the network. However, this is basically what GraphWave does. Thus, if anything, this is a validation of the robustness of the strategy (i.e. STRING+GraphWave vs HINT+HotNet2), but does not really support that the identified genes are related to AD. It would be interesting to benchmark the strategy using as starting point a network not functionally biased and without any literature curation (i.e. HuRI-III; Luck et al. Nature 2020), and checking whether the identified genes are still related to AD.

Conceptually speaking, both EA and PPh2 quantify the likelihood of a protein to be functional in front of a given mutation. Indeed, the most severe mutations often imply the misfolding of the protein or a missing key residue in the active side, and thus a complete lack of function. Accordingly, when assessing protein-protein interactions, if one of the proteins is dead, the interaction (being it physical or functional) will not happen. Thus, I wonder whether using the maximum PS to weight an interaction, or establishing a PS threshold above which the interaction is dead, would be a better option than using the sum of the PS in both proteins. Could this improve the results?

The authors did an incredible work on providing extensive and orthogonal levels of evidence for all the reported different gene lists (e.g. obtained from using different PS scores, cohort samples and networks). Therefore, it would be very valuable to provide a final gene list excel file with the union of all the GeneEMBED gene sets. Then, they can include different columns covering all sources of evidence reported (e.g., the presence of the gene in each PS, cohort and network models, the dysregulation in AD tissues, the overlap with known gene sets, the validation in vivo, etc), and even rank the genes according to the (normalized) degree of evidence provided by this work. This will beautifully summarize the findings reported in the manuscript and will make more evident the identification of potential proteins to be studied.

Throughout the manuscript, the authors draw conclusions regarding the "accuracy" that seem slightly over claimed, as they are based on statistically significant yet modest overlaps or are drawn from not obvious associations. More specifically:

* Lines 163-164: "These data indicate that while GeneEMBED recovers genes that are significantly linked to AD at similar rates as the current state of the art, it does so at smaller cohort sizes". → According to Table S3, this similar rate only applies for the DisGeNet gene set. The other 3 tested gene sets are significantly recovered by MAGMA but barely or not recovered by GeneEMBED counterparts. The suggested sentence seemed to overlook this, generalizing the results obtained in few of the gene sets - gene list comparisons.

* Lines 167-139: "Overall, GeneEMBED identifies candidates distinct from MAGMA which are nonetheless enriched for known AD-associated genes, suggesting an accurate recover of disease signal" → While this recovery "exists" from a statistical point of view (although with modest p-values), the number of overlapping genes is not high enough to be considered an "accurate recover". Indeed, even when taking the highest overlap among the GeneEMBED versions, it does not cover more than 5% of any of the gene sets, with exception of the ClinVar dataset, where the "EA ext" model managed to recover 9.5% of the genes.

* Lines 199-201: "These data suggest that the Gene EMBED candidates are functionally and significantly connected to previously curated AD-associated genes and potential act through similar pathways" → Considering that this analysis is based on the STRING network, which includes multiple sources of evidence (e.g., co-occurrence, experimental, text mining, database, co-expression), the assumption of the participation of the genes in similar pathways cannot be stated from this analysis. In other words, while proteins in similar pathways are included in STRING, unless one limits the analysis to those pathway specific interactions, an association to similar pathways cannot be directly attributed.

* Line 221: "The significant overlap in GeneEMBED candidate genes observed across cohorts and networks..." → Statistical tests showing the significance of the overlaps are reported across cohorts but not across networks. Indeed, Figure S1 shows that the number of overlapping genes across different networks is smaller than the obtained between cohorts.

Minor points and typos

* Line 59-62: "Theoretical analysis suggests that under reasonable assumptions nearly 500,000 samples would be needed to identify statistically significant genetic interactions (16). This motivation has led to the

development of network informed gene prioritization methods for various diseases" → This connection is not clear to me and it is not justified later on in the text. Do the authors mean that network analysis compensates the lack of samples by using prior knowledge?

* Line 88 "...with 481 AD+ and 488 AD-individuals". → Missing space between AD- and individuals.

Reviewer #2: The authors present a new approach to study the effects of variants in the context of gene/protein interactions, which is different from the conventional approaches where variant or gene associations with a phenotype are studied in isolation. Thus the suggested method addresses a gap in the field. By applying this method on two LOAD datasets, using two different variant effect scores and three different networks, the authors show that the method i) can re-discover genes associated with AD, ii) can find new candidates, iii) is robust to lower sample sizes. The software is already openly available on Github and installation using a Conda env & usage is sufficiently described. Application of the method to LOAD datasets yielded the discovery of several candidates, for which the authors provide corroborative evidence using literature or disease databases, disease models, and other omic data. I found the text well-written and easy to understand.

I have a few comments/questions concerning the way comparisons across different methods/scoring measures/datasets/networks are done and the dependence of the method on the network structure. Overall, I am positive about the publication of this study after addressing the questions raised.

Major points:

1- Use of hypergeometric tests where the two methods share data, network, or calculation method is likely not accurate - thus seeing a significant p-value could be misleading. e.g. When the authors compare gene sets obtained using Extension cohort & same network but using different scores, you would already expect a high overlap even if the scores result in low concordance, since the calculations share the network and data, while the p-value is calculated assuming independent sets. I suggest adopting a permutation scheme. E.g. the authors can mix the labels of disease & healthy, calculate embeddings, calculate overlaps, get the distributions and see where their observation lie. This way beyond the p-value, they can also provide the mean number of overlaps in a randomised comparison, which would be a measure of the number of false positives.

2- Although the authors provide a qualitative assessment of the robustness of results by using two different networks, it is important to consider that even networks supported by computational predictions are prone to research bias. Indeed the supplementary analysis suggests a huge influence of network. However, the authors do not provide any measure of their sensitivity to FN and FP edges in the network (e.g. synthetic deletions or additions of edges to see how much the power to detect a difference between case vs. control changes), and do not acknowledge this potential bias in their discussion.

3- The in vivo analysis involves measuring the neuronal dysfunction as an outcome, not directly amyloid or tau aggregation. While I can see the association, I cannot see the exclusivity of the cause-effect relationship between motor performance and B42 or tau accumulation, i.e. change in motor activity could be caused by other factors. Indeed, not all the hits were associated with aggregation-induced neuronal dysfunction. I would suggest authors either explain how this relationship is exclusive or change the emphasis.

4- The authors should give information about the computational efficiency/requirements of the method.

Minor points:

5- Line 168: the authors suggest their gene set is enriched in known AD-associated genes. However, they do not provide an enrichment test - rather they search for the resulting genes in the literature in association with AD and we do not really know if this number of hits would be expected by chance. I suggest rewording to prevent misleading suggestions (this is before CTD, GWAS, DGN, ClinVar analysis)

6- Throughout the text, there are many pairwise comparisons between different results to assess the overlap. It is very difficult to trace them in the text. I suggest the authors consider having a heatmap of effect sizes or p values to clearly demonstrate the results.

7- I would appreciate having more information on overexpression & loss-of-function alleles used in Drosophila and also more experimental details.

8- The quality of supplementary images were quite bad in the referee's copy.

Reviewer #3: In this work, Lagisette et al describe a gene identification method by assessing network properties of genes and genetic observations on two independent international AD cohorts. It is innovative by allowing to compute simultaneous impact of genetic variants with an unbiased approach, to create a "personalized functional impact network" which is important in understanding not only AD but also any other complex disorder. They provide supportive evidence for candidate genes identified by GeneEMBED by comparison with the literature on known AD genes, expression data on post-mortem AD brains and in two different model organisms.

Comments:

- The authors use two different variant impact scores: EA and PPh2. These scores seem to be concordant with 34 overlaps in Discovery cohorts and 44 overlapping genes in Extension cohorts. However, according to Figure 1B, only 4 genes seem to overlap between EA and PPh2 when we compare genes from Discovery+Extension cohorts. Even though it is expected to have lower number of overlaps, did the authors check if this is still significant? I believe this is important to show if one can use GeneEMBED using scores from only one tool or if they need to combine multiple scoring systems as done here. Also, given that there are numerous prediction tools available it is not clear to me if another combination would produce similar results or not. I think this is important to check or discuss.

- The authors download different datasets and perform quality control; therefore, I assume they downloaded the unfiltered datasets. They mention that false-positive variants are removed which I believe by HWE. However, they do not mention if any variant sites are removed by missingness or differential missingness between cases and controls which can impact results in genetic studies. They account for missingness at the sample level; however, this should be applied to variant sites if their dataset was not filtered.

- Selection of "143 high confidence" candidate genes was not clear to me. It is described as "candidate genes identified in at least two independent conditions" referring to Supplementary Figure 1, however a clearer explanation would be helpful for the readers.

- The authors checked if this method would be successful with smaller cohort sizes by down-sampling and found that their tool produce similar results with smaller sizes. However, the smallest sample size they present with high recall rate is 40% of their initial cohort (as presented in Supplementary) which is 2068 samples which is much higher than their extension cohort. So, how does this reflect on the reliability of the findings of their extension cohort? I was very excited about the applicability of this method to smaller cohorts that are also comparable to their extension cohort. However, after seeing this result I'm not sure how easily it can be concluded that this is applicable for smaller cohorts as well.

- It seems like there is a problem with formatting of the references in the Materials and Methods - Whole Exome/Genome Sequencing data.

- In Figure 2 legend, one of the brain regions, PHG, is missing.

- Figure 3, all axis labels should be described in the legend.

Authors' response to the first round of review

Response to the reviewers:

We thank the three reviewers for their thorough and thoughtful comments on our manuscript. We are grateful for the many overall positive comments from all three reviewers on the quality, novelty, and impact of our study. We also understand their remaining questions and agree with the suggestions they

offered. All of these comments have now been incorporated into the manuscript as either clarifications or new data that together address every single one extensively and fully.

As a result, we believe this major revision addresses all the concerns that were raised in the review and as a result is substantially improved. We hope you will agree and that it will now meet the standards expected by the broad readership of Cell Genomics.

We include below our detailed, point-by-point responses to the reviewers' comments. In our efforts to address the reviewers, we have generated two new figures, ten new tables, and revised the text throughout the manuscript.

For clarity:

Comments from the reviewers are in Times italics, black. Our responses to the reviewers are in Arial, black, indented. *Our changes to the text are Arial italic, teal, indented.*

Reviewer #1 wrote:

...The m/s is well structured and clearly written and, while most of the methodology used in the approach is already published, the proposed strategy is novel and original. Furthermore, the authors showed it can be used on small sample sizes, an important limitation for the study of many diseases such as Alzheimer, providing results that were more robust than other state-of-the-art methods. Thus, overall, I am happy to support its publication.

We thank R1 for these positive comments on the writing, the novelty of our strategy, and its robust performance even on small sample sizes, which is usually a limitation of competing techniques. However, there are a few points that should be addressed and clarified, to improve the reproducibility of the work and increase its impact. In particular, I found difficult to follow the strategy used to embed the networks and derive the AD related genes. Additionally, even though the reported results are statistically significant, some of the conclusions seem slightly over optimistic regarding the accuracy of the method. Please, find below some specific comments that, I hope, might improve the clarity and robustness of the m/s. As stated above, the methods section referring to the embedding pipeline requires some clarification, and it also misses important technical details that required for a complete understanding of the methodology. More specifically:

Response to Reviewers

1- Given that GraphWave embeddings are obtained separately for the disease and healthy cohorts, it is not evident if their dimensional spaces can be directly compared (i.e. whether each dimension in the embedding spaces capture equivalent features) since, as far as I know, GraphWave is not an inductive embedding method.

Response: Following the reviewer's suggestion, we have expanded the text to further explain why GraphWave embeddings of disease and healthy graphs can be directly compared. Lines 503 – 510 (manuscript page 21) now reads:

"The GraphWave algorithm has advantages over other embedding algorithms in that it provides rigorous mathematical guarantees on identifying structure preserving embeddings. GraphWave performs unsupervised node embedding on node structure (i.e. topological patterns of node connectivity). Accordingly, Donnat et al provide proof for the equivalency of embeddings between two structurally identical nodes a and b, which rests on the assumption that there exists a one-to-one mapping between the K-hop neighborhood of the two nodes. We can extend this proof to claim that the embeddings of a node from two identical graphs must also be equivalent since there will exist a mapping between the node neighborhoods. Thus, when comparing disease and healthy graphs wherein the node connectivities are largely unchanged, the descriptive features captured by each dimension in the embedding space are the same, thus allowing for direct comparisons. The GraphWave algorithm is briefly described below."

2- The authors justify the use of the PCA with the following state: "This step is necessary because embeddings generated from GraphWave represent complex functions rather than a point in a multidimensional space". However, according to the GraphWave implementation provided by the original authors in their github (<https://github.com/snap-stanford/graphwave>), the algorithm already provides embeddings in the Euclidean space through the characteristic function. Thus, it is not clear which is the

role of the PCA in the methodology as the given justification does not agree with the implementation reported by the GraphWave authors. It would be interesting to see if there are significant differences between distances calculated from the principal components and those extracted from full dimensional embeddings.

Responses: We thank the reviewer for pointing out this apparent inconsistency. We realize our previous justification was not sufficient in explaining our motivation for use of the PCA step and have therefore rewritten the text in lines 538 – 546 (manuscript page 23) to read:

“The role of PCA in the methodology is to aid in denoising the full dimensional embeddings retrieved by GraphWave. The full dimensional embeddings produced by the algorithm can encompass signals ranging from immediate neighborhoods to the complete graph. As a result, the full dimensional embedding of a node will be influenced by any change in the edge weight anywhere in the graph. In order to remove some of these noisy influences, we perform a PCA on the full dimensional embeddings and use the first principal component to measure distances as this component recovers between 78-92% of the variability explained. Characterization of performance between distances computed on full dimensional embedding against distances measured on PCA is shown in sup tables 13-15”

We also thank the reviewer for the valuable computational experiment suggested. In response, we have conducted the experiment suggested by comparing the performance of distances calculated using full dimensional embeddings to those calculated from the principal components. We found that genes identified by PCA were more functionally connected to known AD genes and were significantly enriched for differentially expressed genes from AD brains. We also found that while ~74% of genes identified by full dimensional embeddings were also identified by PCA, genes unique to full dimensional embeddings seemed unrelated to AD suggesting that PCA may act to denoise raw GraphWave outputs. We have added this text in supplemental information lines 50 – 72 (supplemental page 2) and sup tables 13-15:

“Characterization of performance of PCA vs full embedding distances. In order to assess the effects of PCA on the GeneEMBED methodology, we tested the utility of computing distances based on the full dimensional embedding outputs from the GraphWave algorithm compared to PCA-distances. We ran GeneEMBED with both weighting metrics on the Discovery-VISEA cohort using the STRING network and recovered 84 genes with full embedding distances and 69 with PCA-distances. To test the relevance of the identified gene to AD, we measured their: (i) recovery of AD-associated genes, (ii) connectedness to known AD-associated genes, and (iii) differential expression in postmortem AD brains. We found that distances based on full dimensional embeddings were able to recover statistically significant overlaps with GWAS Meta 1, Gwas Meta 2, and DisGeNet. PCA-distance gene set recovered significant overlaps with DisGeNet and CTD (sup table 13). Next, we found that genes identified by full dimensional embeddings were significantly connected to GWAS Meta 1, GWAS Meta 3, and CTD gene sets. Comparatively, PCA-distance gene set showed significant connectivity to all five reference gene sets (sup table 14). Finally, genes identified by full dimensional embeddings showed no statistically significant enrichment for differential expression in post-mortem AD brains, while PCA-distance gene set was enriched in two brain regions with significance ($p = 0.012$) (sup table 15). These data show that distances based on PCA performs better than full dimensional embeddings. Further examination of the genes identified by both approaches showed that 74% of genes identified by the full dimensional embeddings were also identified in the PCA-distance framework. However, of the 20 genes unique to the full dimensional embeddings’ gene set, only 3 were dysregulated in AD brains. Comparatively, of the 7 genes unique to the PCA-distance framework, 4 were dysregulated in AD brains. Overall, this demonstrates the role of principal component analysis in denoising the raw outputs of the GraphWave algorithm.”

3- Along the same line, the authors do not specify the number of PCA dimensions used to calculate the distances (i.e. the number of dimensions/features used to calculate distances between embeddings).

Besides, the variability explained by the used dimensions should be also reported as it helps quantify the information that was eventually used from the learnt GraphWave embeddings.

Response: Yes, we agree. Following R1's suggestion, we have clarified the usage of PCA in the text, which now reads lines 542 – 544 (manuscript page 23):

“In order to remove some of these noisy influences, we perform a PCA on the full dimensional embeddings and use the first principal component to measure distances as this component recovers between 78-92% of the variability explained.”

4- It is not clear how the authors derived a FDR value from the distance distribution. FDR correction is usually used to correct p-values but it seems it was directly implemented, somehow, to the PCA-distances.

This step would also require further clarification.

Response: We agree with the reviewer's point and following this suggestion, we have now clarified the FDR correction procedure. Edited text in lines 546 – 551 (manuscript page 23) now reads:

“By defining distances as the square root of the L2-norm of each gene measured between case and control, we are able to reconstruct a gaussian-like distribution from the positive and negative values. We then compute z-scores and their corresponding p-values for each distance value relative to the full distribution. Then, we perform false discovery rate (FDR) corrections on the p-values using the Benjamini-Hochberg method and genes corresponding to distance values passing $FDR < 0.01$ are selected as pre-candidate genes.”

STRING networks are always dangerous when it comes to interpretation since they are built from many different types of data, including text mining and functional associations. Thus, validating results obtained using STRING networks through reported functional associations (i.e. AD-related genes) is tricky, since this information could have been included in the initial network. Similarly, the authors validate their embeddings strategy by comparing it to a network built from HINT, applying a heat-propagation algorithm (i.e. HotNet2) and computing the distances between nodes in the network. However, this is basically what GraphWave does. Thus, if anything, this is a validation of the robustness of the strategy (i.e. STRING+GraphWave vs HINT+HotNet2), but does not really support that the identified genes are related to AD. It would be interesting to benchmark the strategy using as starting point a network not functionally biased and without any literature curation (i.e. HuRI-III; Luck et al. Nature 2020), and checking whether the identified genes are still related to AD.

Response: We thank the reviewer for raising these relevant points. We recognize the potential and unavoidable research bias that may exist in curated PPI networks. To address the reviewers concern, we performed the suggested experiment using as the input network for GeneEMBED, the unbiased HuRI network (PMID:32296183). This network is the largest unbiased interactome map of binary PPIs to date, containing 8,275 nodes and 52,569 edges. We found that genes identified by using this network recovered no significant overlap or functional connectivity with known AD reference genes.

Candidate genes showed enrichment for differentially expressed genes in AD brains with marginal significance ($p = 0.06$). Though it would seem that GeneEMBED is unable to generate meaningful candidate genes with this unbiased network, it is important to consider the disease context. The HuRI network is an impressive feat of high-throughput analysis, however technological limitations of the field make it an inappropriate network for AD. Specifically, several genes which are fundamental to AD pathobiology, such as APOE, TREM2, and MAPT are nonexistent in the HuRI network. As a result, we would suggest to potential readers to prioritize the relevance a desired network to their disease context. However, if it is possible to use an unbiased network, this would be recommended. We have made the following changes in lines 403 – 411 (manuscript page 17):

“We found that GeneEMBED consistently identified similar genes across the three PPI networks used in this study ($p \sim 1.06e-8 - 5.78e-28$, sup fig. 2), suggesting that usage of any well-constructed and disease relevant network will tend to converge on similar findings. While the use of networks is key in the GeneEMBED strategy, it also introduces a potential source of error. Even stringently curated networks may be prone to research bias. Unbiased networks built with high through-put techniques may provide alternatives. However, they tend to be limited in size due to technical constraints, resulting in an insufficient capture of disease relevant interactions (sup materials, sup tables 16-18). In this regard, the

GeneEMBED strategy showed robustness to the presence of both false positive and false negative edges (sup materials, sup fig 7).”

We have also detailed this information to supplementary materials in lines 137 – 161 (supplemental page 5):

“Characterization of GeneEMBED performance in unbiased networks. In order to benchmark the GeneEMBED strategy with a network without any functional bias or literature curation, we employed the HuRI network. The HuRI network is the largest unbiased interactome map of binary protein-protein interactions. The network contains 8,275 nodes and 52,569 edges generated from an impressive array of nine different ‘all-by-all’ screens of 17,408 proteins.

Using this network as a starting point, we ran GeneEMBED using the Discovery- VISEA cohort and identified a candidate gene set. To test the relevance of the identified genes to AD biology, we examined: (i) direct overlaps with reference gene sets discussed previously, (ii) connectedness between reference gene sets and identified genes, and (iii) dysregulation of identified genes in postmortem AD and non-AD samples from the AMP-AD dataset. Performing these experiments, we found that there was no significant recovery of known AD-associated genes. We also found no significant preferential connectivity between candidate genes and known AD-associated genes (sup tables 16, 17). We did find an enrichment of the candidate genes for differentially expressed genes in AD vs non-AD brains with marginal significance ($p = 0.06$) (sup table 18). While these results would seem to suggest that GeneEMBED is unable to perform on such unbiased networks, it is important to consider the HuRI network in the context of AD. Despite being the largest of its kind to date, due to technological limitations, the HuRI network comprises only half the exome. Accordingly, only half or less of the genes in the reference gene sets were present in the HuRI network. Indeed, several genes which are core to AD pathobiology, such as APOE, TREM2, or MAPT, were absent in HuRI. The stringency of the HuRI network’s construction suggests that while it has a low FP rate, it may be depleted in protein-protein interactions. Indeed, we have observed that GeneEMBED is more robust to FP edges than FN edges (sup materials, sup fig 7). Overall, these data emphasize the importance of appropriately selecting a starting network. While it is recommended to use an unbiased network when possible, it is also crucial to ensure the network is reflective of the biology of the target disease.”

Conceptually speaking, both EA and PPh2 quantify the likelihood of a protein to be functional in front of a given mutation. Indeed, the most severe mutations often imply the misfolding of the protein or a missing key residue in the active side, and thus a complete lack of function. Accordingly, when assessing protein-protein interactions, if one of the proteins is dead, the interaction (being it physical or functional) will not happen. Thus, I wonder whether using the maximum PS to weight an interaction, or establishing a PS threshold above which the interaction is dead, would be a better option than using the sum of the PS in both proteins. Could this improve the results?

Response: The reviewer raises a good point. Following the suggestion, we characterized GeneEMBED performance using these alternative edge weighting schemes. We found that the use of the maximum PS ($\max(\text{PS})$) or the PS threshold did not perform as well as the current implementation of $\text{sum}(\text{PS})$ when considering recovery of known AD genes, preferential connectivity to known AD genes, or enrichment for differentially expressed genes in AD brains. We propose that this is because $\text{sum}(\text{PS})$ provides extra resolution in characterizing perturbed edges that the other two schemes do not. If we consider a protein with many physical interactions, an impactful mutation in some active site may induce a high EA or PPh2 score. This would mean that any interaction mediated by this active site will be perturbed, but it does not necessitate that all other interactions at other sites along the protein will be perturbed to the same degree. $\text{Sum}(\text{PS})$ allows this extra resolution which leads to improved performance.

We have included the following text changes to lines 412 – 419 (manuscript page 17):

“Namely, the edge weighting scheme. While other edge weighting approaches were characterized (sup materials, sup tables 13-15), the current framework estimates the perturbation of each interaction independently but considers all edges equally important. However, biological networks are highly robust to mutations due to pathway redundancies (125,126). Among these, some are dominant while others are

auxiliary (127), suggesting that different parts of the network have varying levels of importance. This indicates a potential limitation and area for improvement in the GeneEMBED framework. Potential approaches to address this are to consider alternative methods of node embedding including anisotropic diffusion techniques, which will be the focus of future work.”

We have included in supplementary information the lines 74 – 104 (supplemental page 4):

“Characterization of alternative edge weighting schemes. In order to characterize the performance of the GeneEMBED methodology to differences in edge weighting, we adopted two alternative weighting schemes. First, we implemented a max(PS) scheme wherein the weight of a given edge is simply determined by the maximum PS of the two connected genes. Second, we applied a PS threshold ($PS > 0.7$), wherein if either of the connected genes has PS above the threshold then the interaction is considered dead. Otherwise the weight is determined by the maximum PS between the two genes. We ran GeneEMBED with both weighting metrics on the Discovery-VISEA cohort using the STRING network and compared outputs with the original framework which uses sum(PS). We recovered 73 genes from max(PS) and 72 genes from PS threshold. Interestingly, nearly 90% of identified genes overlapped with the sum(PS) based gene set. To test the relevance of the two gene sets to AD, we measured their: (i) recovery of AD-associated genes, (ii) connectedness to known AD-associated genes, and (iii) differential expression in postmortem AD brains. Both max(PS) and PS threshold schemes recovered statistically significant overlaps in the DisGeNet reference gene set (sup table 13).

Comparatively, the current weighting scheme, sum(PS), recovered similar statistically significant overlaps in DisGeNet as well as significant overlaps with the CTD gene set. Next, max(PS) showed significant diffusion to ClinVar and GWAS Meta 1 gene sets, while PS threshold showed significant diffusion to all gene sets except CTD (sup table 14). Comparatively, sum(PS) showed statistically significant diffusion to all gene sets. Lastly, we found that genes identified by max(PS) were enriched for differential expression in one brain region out of seven, though this was not significant ($p = 0.076$). The genes identified by the PS threshold scheme, like sum(PS), were enriched for differentially expressed genes in two brain regions ($p = 0.0014$ and 0.012 respectively) (sup table 15). These data suggest that though the three weighting schemes overlap in close to 90% of their identified genes, sum(PS) performs better than alternative weighting methods. This is because sum(PS) allows each interaction to be described independently. For example, consider a protein with many interactions. If the protein has a mutation in some binding site, this may lead to a high PS due to the inactivation of the binding site. In this case, the interaction between the protein and any interactor which is mediated by the mutated binding site is significantly perturbed. However, this does not necessarily mean that the protein’s interactions mediated by other unaffected binding sites are perturbed to the same degree. Sum(PS) allows for this flexibility while the other weighting schemes do not, which leads to its improved performance.”

We have also added the following sentence to lines 494– 495 (manuscript page 21):

“Characterization of alternative edge weighting schemes and their corresponding discussions can be found in supplementary information and sup tables 13-15.”

The authors did an incredible work on providing extensive and orthogonal levels of evidence for all the reported different gene lists (e.g. obtained from using different PS scores, cohort samples and networks). Thank you. Therefore, it would be very valuable to provide a final gene list excel file with the union of all the GeneEMBED gene sets. Then, they can include different columns covering all sources of evidence reported (e.g., the presence of the gene in each PS, cohort and network models, the dysregulation in AD tissues, the overlap with known gene sets, the validation in vivo, etc), and even rank the genes according to the (normalized) degree of evidence provided by this work. This will beautifully summarize the findings reported in the manuscript and will make more evident the identification of potential proteins to be studied.

Response: We completely agree and thank the reviewer for this excellent thought. We have generated such a summary table and have incorporated it into the text in lines 228 – 232 (manuscript page 10) and sup table 21:

“These genes were selected using the criteria that they must have been identified at least twice in the same network either across cohorts or across VIS methods. Genes were prioritized based on the degree of overlap across networks with more recurrent genes ranking higher, provided that they were never identified in any of the healthy control vs healthy control assays (sup fig.1, sup table 21)”

Throughout the manuscript, the authors draw conclusions regarding the "accuracy" that seem slightly over claimed, as they are based on statistically significant yet modest overlaps or are drawn from not obvious associations. More specifically:

* Lines 163-164: "These data indicate that while GeneEMBED recovers genes that are significantly linked to AD at similar rates as the current state of the art, it does so at smaller cohort sizes". →

According to Table S3, this similar rate only applies for the DisGeNet gene set. The other 3 tested gene sets are significantly recovered by MAGMA but barely or not recovered by GeneEMBED counterparts. The suggested sentence seemed to overlook this, generalizing the results obtained in few of the gene sets – gene list comparisons.

Response: We thank the reviewer for their comment and have deemphasized the relevant sentence, which now reads lines 167 – 168 (manuscript page 7):

“These data suggest that GeneEMBED is able to significantly recover several known AD genes despite large differences in cohort sizes.”

* Lines 167-139: "Overall, GeneEMBED identifies candidates distinct from MAGMA which are nonetheless enriched for known AD-associated genes, suggesting an accurate recover of disease signal" → While this recovery "exists" from a statistical point of view (although with modest p-values), the number of overlapping genes is not high enough to be considered an "accurate recover". Indeed, even when taking the highest overlap among the GeneEMBED versions, it does not cover more than 5% of any of the gene sets, with exception of the ClinVar dataset, where the "EA ext" model managed to recover 9.5% of the genes.

Response: We agree with the reviewer and following their suggestion, we have tempered our characterization of GeneEMBED's performance as accurate recovery. The now reads lines 171 – 173 (manuscript page 8):

“Overall, GeneEMBED identifies candidates distinct from MAGMA which are nonetheless enriched for known AD-associated genes, suggesting an identification of disease relevant signal.”

* Lines 199-201: "These data suggest that the Gene EMBED candidates are functionally and significantly connected to previously curated AD-associated genes and potential act through similar pathways" → Considering that this analysis is based on the STRING network, which includes multiple sources of evidence (e.g., co-occurrence, experimental, text mining, database, co-expression), the assumption of the participation of the genes in similar pathways cannot be stated from this analysis. In other words, while proteins in similar pathways are included in STRING, unless one limits the analysis to those pathway specific interactions, an association to similar pathways cannot be directly attributed.

Response: We thank the reviewer for their comment and understand their concerns. We have deemphasized the relevant sentence, which now reads lines 203 – 205 (manuscript page 9):

“These data show that the GeneEMBED candidates are significantly and functionally connected to previously curated AD-associated genes, further suggesting an identification of disease relevant signal.”

* Line 221: "The significant overlap in GeneEMBED candidate genes observed across cohorts and networks..." → Statistical tests showing the significance of the overlaps are reported across cohorts but not across networks. Indeed, Figure S1 shows that the number of overlapping genes across different networks is smaller than the obtained between cohorts.

Response: We thank the reviewer for pointing out this fact. To address the reviewer's concern, we have provided a heatmap of hypergeometric overlap p-values across cohorts, networks, and VIS methods. This information has been included as supplementary material sup figure 2:

Figure Legend: GeneEMBED candidates are consistently identified across various cohorts, networks, and VIS systems. Hypergeometric overlap tests were done on every pairwise combination of cohort-network-VIS experiments. Among 66 independent pairwise tests, only 11 did not demonstrate statistically significant hypergeometric p-values ($p < 0.05$, $\log(p) < -2.99$).

Minor points and typos:

* Line 59-62: "Theoretical analysis suggests that under reasonable assumptions nearly 500,000 samples would be needed to identify statistically significant genetic interactions (16). This motivation has led to the development of network informed gene prioritization methods for various diseases" → This connection is not clear to me and it is not justified later on in the text. Do the authors mean that network analysis compensates the lack of samples by using prior knowledge?

Response: Yes, we agree with the reviewer's point and following their important request for clarification, we have now included in the text lines 59 – 62 (manuscript page 3):

"Theoretical analysis suggests that under reasonable assumptions nearly 500,000 samples would be needed to identify statistically significant genetic interactions (16). The potential use of prior knowledge to compensate for necessary sample size has motivated the development of network informed gene prioritization methods for various diseases (23–28)"

* Line 88 "...with 481 AD+ and 488 AD-individuals". → Missing space between AD- and individuals.

Response: We thank the reviewer for nothing this and have revised lines 89 (manuscript page 4):

"...with 481 AD+ and 488 AD- individuals.."

Reviewer #2 wrote:

...I found the text well-written and easy to understand. I have a few comments/questions concerning the way comparisons across different methods/scoring measures/datasets/networks are done and the dependence of the method on the network structure. Overall, I am positive about the publication of this study after addressing the questions raised.

We thank Reviewer 2 for these positive comments regarding the overall study as well as the narrative structure.

Major points:

1- Use of hypergeometric tests where the two methods share data, network, or calculation method is likely not accurate - thus seeing a significant p-value could be misleading. e.g. When the authors compare gene sets obtained using Extension cohort & same network but using different scores, you would already expect a high overlap even if the scores result in low concordance, since the calculations share the network and data, while the p-value is calculated assuming independent sets. I suggest adopting a permutation scheme.

E.g. the authors can mix the labels of disease & healthy, calculate embeddings, calculate overlaps, get the distributions and see where their observation lie. This way beyond the p-value, they can also provide the mean number of overlaps in a randomized comparison, which would be a measure of the number of false positives.

Response: We thank the reviewer's thoughtful and keen comments regarding the use of hypergeometric tests in these cases where the two compared sets are not independent of one-another. We also appreciate the experimental framework suggested by the reviewer of a permutation scheme to define a background of overlaps. While the suggestion is astute, the integrative nature of the GeneEMBED approach makes this sort of experimental scheme inappropriate. Shuffling labels between disease and healthy will lead to noisy and uninformative mutational input. This will cause GeneEMBED to heavily rely on network data and identify differentially mutated genes between shuffled labels based on noise alone. These genes may have some network topological characteristics which boost small and noisy signals between the two labeled groups. Such genes may be potential false positives (FPs) and can also be identified in healthy control vs healthy control analyses. Therefore, in the original GeneEMBED framework, we performed control vs control analyses across all network and VIS modalities and considered all significantly identified genes to be potential FPs. These FPs were removed from any proceeding case vs control analyses. Given that the FPs relating to shared network or VIS data were removed, candidate gene sets identified from the Discovery or Extension cohorts were considered independent. Thus, hypergeometric overlap tests were performed on these post-processed, independent candidate gene sets.

This idea has been described with modifications in lines 126 – 127 (manuscript page 6):

“Additionally, we applied GeneEMBED to healthy control vs healthy control using both VIS_{EA} and VIS_{PPH2} to identify potential false positive (FP) genes.”

We have also further modified lines 551 – 556 (manuscript page 23):

“Lastly, the full GeneEMBED process is performed on healthy controls vs healthy controls (details of healthy control selection are discussed below). Genes passing $FDR < 0.01$ threshold in this control vs control analysis are removed from the list of pre-candidate genes. This is done to filter potential sources of variation which may not be disease specific (false positives, sup materials). The final set of genes passing $FDR < 0.01$ threshold and not removed by control vs control analysis are considered candidate genes.”

In order to further address the reviewer's concern, we performed the shuffled label analysis as suggested. We demonstrate that when mutational data is noisy and counterproductive, GeneEMBED relies heavily on network data with prioritized genes correlating significantly with network connectivity. Furthermore, we demonstrate that overlapping genes identified from shuffled label trials are far less associated with AD than genes that were identified in case vs control analyses. These data have been detailed in supplementary information in lines 163 – 201 (supplemental page 6):

“Characterization of GeneEMBED in the presence of uninformative mutational data. In the presence of counterproductive mutational data and large influence from network inputs, similar genes will be recovered from various shuffled label experiments leading to inflated overlaps. Indeed, it is likely that due to ambiguous mutational input the identified overlapping genes from randomly shuffled trials are less related to AD than case vs control overlaps.

Moreover, the large reliance on network information suggests that identified gene lists are strongly correlated with network connectivity. In order to test these hypotheses, we performed the shuffled labeled experiments in the STRING network for both VIS_{EA} and VIS_{PPH2} using Discovery and Extension cohorts. We found that for VIS_{EA} , an average of 31.7 genes overlapped among gene sets identified using label shuffling for the Discovery and Extension cohorts.

Comparatively, an overlap of 14 genes was observed in the original framework after removing potential FPs from control vs control analysis. Importantly, we found that the 14 genes identified in the original analysis showed significant hypergeometric overlap with all five of the reference gene sets of known AD-associated genes ($p = 0.019 - 0.0039$). The overlaps identified by shuffled labeling showed few to no significant overlaps with any of the five reference gene sets (sup table 19). Next, for each of the 14 genes

in the original analysis, we counted the number of publications in the PubMed database co-mentioning the gene with AD in abstracts. We randomly generated 50 gene sets of the same size and counted their co-mentions with AD to build a background distribution. To determine if a gene was related to AD, we used a threshold of at least 5 publications co-mentioning a gene and AD. We found that relative to this background, the original observation of 14 overlapping genes had a z-score of 6.86. Comparatively, the overlaps identified by random shuffling had an average z-score of 3.43 and stdev of 1.57. Lastly, we found that ranked gene lists derived from random shuffling were significantly correlated with degree centrality (pearson correlation coeff. = 0.2-0.36, $p = 0.037 - 1.7e-5$), whereas the gene list derived from case vs control analysis was not correlated with pearson correlation coefficient of 0.085 and a p value = 0.43. Similarly, in VISPPH2 analysis, an average of 37.5 genes overlapped among gene sets identified by label shuffling. In contrast, 16 genes were found overlapping between Discovery and Extension cohorts using the original framework after removing potential FPs. While no significant overlaps were observed with the reference gene sets (sup table 20), we found that the PubMed literature curation analysis showed significant association of overlap genes identified from case vs control analysis to AD with a z-score of 4.49. In comparison, overlaps obtained from label shuffling had an average z-score of 0.97 and stdev of 0.82.

These data suggest that overlaps observed between Discovery and Extension cohorts in the original analysis are much smaller than expected by label shuffling trials. Despite these large differences in sizes, overlaps from the original framework are more related to AD. Further, they tend to rank genes independently of their pure connectivity, whereas label shuffling leads to a heavy dependence on network information. Overall, these observations demonstrate that during a lack of informative mutational data, GeneEMBED will tend to depend heavily on network information, identifying genes which are less relevant to AD than genes identified through productive (case vs control) mutational data.”

2- Although the authors provide a qualitative assessment of the robustness of results by using two different networks, it is important to consider that even networks supported by computational predictions are prone to research bias. Indeed the supplementary analysis suggests a huge influence of network. However, the authors do not provide any measure of their sensitivity to FN and FP edges in the network (e.g. synthetic deletions or additions of edges to see how much the power to detect a difference between case vs. control changes), and do not acknowledge this potential bias in their discussion.

Response: We thank the reviewer for raising a very relevant point. We have characterized the sensitivity of GeneEMBED to FN and FP edges in the network as suggested by the reviewer. We found that because there are likely far more non-AD-associated genes in the genome than AD-associated genes, the number of edges which significantly influence the recovery of candidate genes are outnumbered by non-influential edges. As a result, we found that up to 55% of edges in the original network could be randomly deleted before either recall or precision dropped below 0.6. However, when we restricted randomly dropped edges to those that involve candidate genes, we found that up to 30% of edges could be deleted before recall or precision fell below 0.6. Conversely, when we randomly added edges, we found that we could add up to ~38.4k edges (80% of the original network size) before precision or recall fell below 0.6. Overall, these data suggest that GeneEMBED is highly robust to large variations in FP and FN edges.

These data have been expressed in the discussion in lines 404 – 411 (manuscript page 17):

“We found that GeneEMBED consistently identified similar genes across the three PPI networks used in this study ($p \sim 1.06e-8 - 5.78e-28$, sup fig. 2), suggesting that usage of any well-constructed and disease relevant network will tend to converge on similar findings. While the use of networks is key in the GeneEMBED strategy, it also introduces a potential source of error. Even stringently curated networks may be prone to research bias. Unbiased networks built with high through-put techniques may provide alternatives. However, they tend to be limited in size due to technical constraints, resulting in an insufficient capture of disease relevant interactions (sup materials, sup tables 16-18). In this regard, the GeneEMBED strategy showed robustness to the presence of both false positive and false negative edges (sup materials, sup fig 7).”

We have also detailed these data and findings in supplementary materials in lines 105 –

135 (supplemental page 4) and sup fig 7:

“Characterization of sensitivity of GeneEMBED to false negative and false positive edges. While the curation of biological networks has become increasingly more sophisticated, it is important to recognize that even networks built upon stringent curation of experimentally validated edges may be prone to research bias. In order to assess the robustness of GeneEMBED to these potential false positive (FP) or false negative (FN) edges, we performed iterative random deletion or addition of edges to the Brain specific PPI network and applied GeneEMBED to the modified network. Using the genes identified by the discovery cohort with the VIS-EA as ground truth, we calculated precision and recall of gene sets identified in the modified networks wherein 5% to 70%, with intervals of 5%, of edges were synthetically and randomly deleted (or added).

Specifically, we used the Brain specific network for this experiment because its relatively small size of 3.2k nodes and 48k edges allowed for computationally efficient modification and testing. First, testing the relationship between FN edges and GeneEMBED performance, we found that up to 55% of the edges in the original network could be randomly deleted before either recall or precision fell below 0.6 (sup fig 7A). Moreover, when we restricted random deletion of edges to those involving any of the genes identified in the unaltered Brain specific PPI network, we found that up to 30% of their edges could be deleted before either recall or precision fell below 0.6 (sup fig 7A). Next, characterizing GeneEMBED performance in the presence of FP edges, we found that we could randomly add edges totaling up to 80% of the original network size (~38.4k edges) before either the precision or recall fell below 0.6 (sup fig 7B). These data suggest GeneEMBED is highly robust to both false positive and false negative edges. In the case of random deletion of edges (FN edges), it is likely that there are more genes that do not play a role in AD pathobiology than genes that contribute significantly to pathogenesis. Accordingly, there will be more edges that are not associated with AD than edges that are associated with AD.

Therefore, it is possible to randomly delete a large number of edges while maintaining a high recall and precision. However, when there is a bias in the edge deletion process to informative edges, the methodology becomes more sensitive to FN edges. Similar reasoning can be applied to the case of random edge addition (FP edges), as there are likely more edges that are not associated with AD it is possible to have large numbers of FP edges before recall or precision drop below 0.6. Overall, these data show that while there may be potential research bias in curated biological networks, the strategy employed by GeneEMBED allows for its robustness to the presence of false positive and false negative edges”

3- The in vivo analysis involves measuring the neuronal dysfunction as an outcome, not directly amyloid or tau aggregation. While I can see the association, I cannot see the exclusivity of the cause-effect relationship between motor performance and B42 or tau accumulation, i.e. change in motor activity could be caused by other factors. Indeed, not all the hits were associated with aggregation-induced neuronal dysfunction. I would suggest authors either explain how this relationship is exclusive or change the emphasis.

Response: We thank the reviewer for their comment and welcome the opportunity to clarify and expand on our rationale for the use of *Drosophila*. Per the reviewer’s suggestion, we have further explained the use of *Drosophila* in the main text in lines 266

– 268 (manuscript page 11):

“Therefore, to optimally validate the GeneEMBED candidates in the AD context in vivo we resorted to *Drosophila* AD models, which capture important core AD traits, including age dependence and protein accumulation (90).”

We have also added lines 273 – 275 (manuscript page 12):

“Expression of secreted β 42 or human tau specifically in post-mitotic neurons induces progressive nervous system dysfunction in *Drosophila* which can be monitored by measuring the motor performance of the animals as they age”

Additionally, we have added the lines 283 – 292 (manuscript page 12):

“Importantly, some of the *Drosophila* alleles used (inducible overexpression and shRNA lines) were targeted specifically to neurons and therefore likely exerted their effects specifically in neuronal cells. However, other alleles used were classical loss of function or classical rescue constructs (using the endogenous gene promoter) in those cases the effect maybe cell-autonomous or non-cell autonomous for example through modulation of important functions in glial or muscular cells. Additionally, while some of the modifiers identified may exert their effect through modulating the accumulation of tau or β 42, others may act by protecting or potentiating the predisposition of neurons to degenerate or even by causing certain levels of neurodegeneration themselves. A complete list of the modifier alleles as well as brief description of their putative effect on their target gene is available in supplemental table 12.”

4- The authors should give information about the computational efficiency/requirements of the method.

Response: We completely agree with the reviewer and following their suggestion, we have added the following lines to the text to expand on the computational efficiency and requirements lines 557– 567 (manuscript page 24):

“Computational efficiency/requirements. GeneEMBED offers an analytical framework to appraise all coding genes in the human genome with respect to their attributes in a molecular network. Accordingly, this can be computationally demanding depending on the size of the network being used. In this study we used three different PPI networks, a brain specific network, HINT, and STRING.

After annotation and preprocessing of exomic variant calling file (VCF), the computational time required for the brain network consisting of 3.2k nodes and 48k edges was 649 seconds (10.8 minutes). Similarly, for the HINT network consisting of 12.6k nodes and 146k edges, the computational time from annotation of networks with mutational information to identification of candidate genes was 3058 seconds (51 minutes). Lastly, for the STRING network consisting of 15k nodes and 1.9m edges, the computational time was 6.2 hours. All network analyses were performed on a server with specifications of Intel Xeon Gold 5222 CPU at 3.8GHz with 8 cores and 348gb RAM.”

Minor points:

5- Line 168: the authors suggest their gene set is enriched in known AD-associated genes. However, they do not provide an enrichment test - rather they search for the resulting genes in the literature in association with AD and we do not really know if this number of hits would be expected by chance. I suggest rewording to prevent misleading suggestions (this is before CTD, GWAS, DGN, ClinVar analysis)

Response: We thank the reviewer for noting this and have revised the sentence accordingly. The text in lines 144 – 146 (manuscript page 7) now reads:

scoring systems and can recover several wellcharacterized, positive control AD genes.”

6- Throughout the text, there are many pairwise comparisons between different results to assess the overlap.

It is very difficult to trace them in the text. I suggest the authors consider having a heatmap of effect sizes or p values to clearly demonstrate the results.

Response: We thank the reviewer for their valuable contribution. We agree with this point and have addressed it along with a similar comment from Reviewer 1 through the addition of a supplementary figure 2.

7- I would appreciate having more information on overexpression & loss-of-function alleles used in *Drosophila* and also more experimental details.

Response: We agree with the reviewer that this would be valuable information for the reader. Following the reviewer’s suggestion, we have expanded on the experimental details in the Materials and Methods section in lines 664 – 685 (manuscript page 28). We have also added calls to supplemental table 12 in the results lines 292 (manuscript page 12). and materials and methods lines 669 (manuscript page 28). This table includes all the alleles tested, as well as descriptions related to their effect on the target gene and the gene correspondence between *Drosophila* and human.

The modified methods in lines 664 – 685 (manuscript page 28) now reads:

“Genetics and strains: *Drosophila* lines carrying UAS-Tau, and UAS-Aos:β42 have been previously described (128,129) and are available from the Bloomington *Drosophila* Stock Center (BDSC, University of Indiana). For post mitotic pan-neuronal expression we used the elav-GAL4(C155) driver from BDSC. The alleles tested as potential modifiers targeting the *Drosophila* homologs of GeneEMBEd candidate genes were obtained from the BDSC.

Homologs were identified using BLAST and also the DRSC Integrative Ortholog Prediction Tool (Diopt score) (130,131)(Supplemental table 12). For the neuronal dysfunction tests, we used a highly automated behavioral (motor performance) assay based on the *Drosophila* startle-induced negative geotaxis response as previously described (131,132). To assess motor performance of fruit flies as a function of age, we used 10 age-matched virgin females per replica per genotype. Four replicates were used per genotype. Flies are collected in a 24-hour period and transferred into a new vial containing 300μl of semi defined media (20g yeast, 20g Tryptone, 30g sucrose, 60g Glucose, 0.5g MgSO₄·7H₂O, 0.5g CaCl₂·2H₂O, 80g Inactive Yeast, 1L H₂O) every day.

Using an automated platform that uses a mechanized arm and clamp (<https://nri.texaschildrens.org/core-facilities/high-throughput-behavioralscreening-core>), the animals are tapped to the bottom of a plastic vials to trigger their negative geotactic response (climbing response) and are recorded for 7.5 seconds as they climb on the walls of transparent plastic vials. Videos are analyzed using custom software (code available for download on ref (131)) that assigns movement trajectories to each individual animal, assesses their speed (mm/s) and returns an average per replicate per trial. Three trials per replicate are performed each day shown, and four replicates per genotype are used. A mixed effect model analysis of variance using spline regressions was run on Rstudio, using each four replicates to establish statistical significance across genotypes (132). Human genes POLD1 and ANLN were identified as modifiers in a separate manuscript currently under revision and were not directly tested here.”

8- The quality of supplementary images were quite bad in the referee's copy.

Response: Quality of the supplementary figures have been refined and upgraded.

Reviewer #3:

In this work, Lagisetty et al describe a gene identification method by assessing network properties of genes and genetic observations on two independent international AD cohorts. It is innovative by allowing to compute simultaneous impact of genetic variants with an unbiased approach, to create a "personalized functional impact network" which is important in understanding not only AD but also any other complex disorder. They provide supportive evidence for candidate genes identified by GeneEMBEd by comparison with the literature on known AD genes, expression data on post-mortem AD brains and in two different model organisms.

We thank Reviewer 3 for their positive feedback regarding the innovative nature of our strategy and its potential use in a wide variety of complex genetic diseases.

Comments:

- The authors use two different variant impact scores: EA and PPh2. These scores seem to be concordant with 34 overlaps in Discovery cohorts and 44 overlapping genes in Extension cohorts. However, according to Figure 1B, only 4 genes seem to overlap between EA and PPh2 when we compare genes from Discovery+Extension cohorts. Even though it is expected to have lower number of overlaps, did the authors check if this is still significant? I believe this is important to show if one can use GeneEMBEd using scores from only one tool or if they need to combine multiple scoring systems as done here. Also, given that there are numerous prediction tools available it is not clear to me if another combination would produce similar results or not. I think this is important to check or discuss.

Response: We thank the reviewer for raising a very relevant point. We admit that significance of the observed overlap across EA and PPh2 when comparing Discovery + Extension cohorts was not tested. We also agree that we had not sufficiently discussed the compatibility of GeneEMBEd with other variant impact tools. We have addressed the first point with changes in lines 143 – 144 (manuscript page 6): “Lastly, we found that 4 genes overlapped among all cohort-VIS combinations with a hypergeometric p-value ~ 8.58e-10.”

We have also made changes to lines 396 – 402 (manuscript page 17) in order to address the second point brought up by the reviewer:

“Here, we used EA due to its consistently good performance in blind, objective studies (121,122) and overall utility in genomic studies (123,124) in addition to a well-established alternative, PPh2. Despite their differences, we found significant overlap in their predictions ($p \sim 2.46e-64 - 1.46e-53$), supporting the compatibility of GeneEMBED with multiple impact estimators. The GeneEMBED framework crucially relies on the PS metric which is compatible only with estimators that have probabilistic interpretations. While some tools (REVEL, SIFT, MutPred2, or VEST) fit this criterion, many do not have such interpretations or may require further transformations (e.g. CADD or Eigen).”

- The authors download different datasets and perform quality control; therefore, I assume they downloaded the unfiltered datasets. They mention that false-positive variants are removed which I believe by HWE. However, they do not mention if any variant sites are removed by missingness or differential missingness between cases and controls which can impact results in genetic studies. They account for missingness at the sample level; however, this should be applied to variant sites if their dataset was not filtered.

Response: We thank the reviewer for raising this comment. We agree that clarification is required with regards to this point. We would like to emphasize that the cohort data we had downloaded were fully filtered and stringently quality controlled (QC) by the ADSP and GCAD consortia, details of which can be found in (PMID: 29590295). In addition to the rigorous QC performed by the publishing consortia, we performed supplemental QC by calculating statistics for Ti/Tv, number of variants, and missingness for each sample as well as HWE and genotyping rate for each variant site across cases and controls. We then filtered out variant sites that were HWE violated as well as variants whose genotyping rate was less than 0.95 in either case or control and in combined case and control samples.

We have clarified these details in the Materials and Methods section, the text in lines 442 – 449 (manuscript page 19) now reads:

“Although extensive QC procedures were performed on the WES Discovery and WGS extension cohorts by the ADSP and GCAD consortia (128), respectively, we generated QC statistics for Ti/Tv, number of variants, singletons, and missingness for each sample and HWE, genotyping rate (AC/AN) for each variant site across cases and controls. Then, potentially false-positive variants sites and outlier samples were removed. HWE (Hardy Weinberg Equilibrium) exact test (129) was performed on the control samples of each cohort and the variants with HWE violations (HWE p -value $< 5E-8$) were removed. We also removed the variants that were genotyping rate less than 0.95 in either case and control and in combined case and control samples.”

- Selection of "143 high confidence" candidate genes was not clear to me. It is described as "candidate genes identified in at least two independent conditions" referring to Supplementary Figure 1, however a clearer explanation would be helpful for the readers.

Response: We thank for noting this point. We have clarified this item in the text which now reads as follows in lines 228 – 232 (manuscript page 10):

“We constructed a network in STRING with 143 high confidence hits. These genes were selected using the criteria that they must have been identified at least twice in the same network either across cohorts or across VIS methods. Genes were prioritized based on the degree of overlap across networks with more recurrent genes ranking higher, provided that they were never identified in any of the healthy control vs healthy control assays (sup fig. 1, sup table 21).”

- The authors checked if this method would be successful with smaller cohort sizes by down-sampling and found that their tool produce similar results with smaller sizes. However, the smallest sample size they present with high recall rate is 40% of their initial cohort (as presented in Supplementary) which is 2068 samples which is much higher than their extension cohort. So, how does this reflect on the reliability of the findings of their extension cohort? I was very excited about the applicability of this method to smaller cohorts that are also comparable to their extension cohort. However, after seeing this result I'm not sure how easily it can be concluded that this is applicable for smaller cohorts as well.

Response: We thank the reviewer for this very relevant comment. We agree that the point at which recall drops below 0.6 for both VIS_{EA} and VIS_{PPH2} is at 40% of the initial cohort size (sup fig 5A). However, we would like to note that though recall of VIS_{PPH2} decreases beyond this point, the recall of the VIS_{EA} is maintained at approximately 0.6. These data would suggest that the robustness of GeneEMBED to very small cohort sizes (e.g. 1000 samples) would rely in part on the VIS tool used. Additionally, while this analysis shows the tentative effect of decreasing sample size, it does not account for hard-to-model effects such as inter-cohort variability. These effects may have large impacts on the quality of identified candidate genes. For example, though the Extension cohort is much smaller, candidate genes identified by VIS_{EA} and VIS_{PPH2} showed more enrichment for differentially expressed genes in postmortem AD brains than the larger Discovery cohort (fig 2A, B).

Lastly, while we validated the AD-association of candidate gene list derived from the Discovery and Extension cohorts independently, we use these cohorts in a complementary manner by compiling the most robustly identified genes across the two cohorts, VIS systems, and networks. In this way we increase confidence in our genes by ensuring that they are robust to variations in sequencing, population structure, VIS tools, and network information. Accordingly, we would recommend potential readers to a similar strategy of validating their candidate genes in two independent cohorts.

We have further emphasized and clarified this point through modifications in lines 393 – 395 (manuscript page 17):

“Nevertheless, in order to optimally account for the various factors leading to inter-cohort variability and increase robustness of findings, we recommend readers to validate potential candidate gene lists across two or more cohorts.”

- It seems like there is a problem with formatting of the references in the Materials and Methods - Whole Exome/Genome Sequencing data.

Response: We thank the reviewer for noting this and have made the necessary formatting changes for references in the Whole Exome/Genome Sequencing section of the Materials and Methods.

- In Figure 2 legend, one of the brain regions, PHG, is missing.

Response: We thank the reviewer for noting this issue and have made the necessary change. The figure legend now reads:

“Figure 2. GeneEMBED candidates are differentially expressed in AD brain tissue. (A) Enrichment of GeneEMBED candidates against differentially expressed genes from seven brain regions: cerebellum (CBE), temporal cortex (TCX), frontal pole (FP), inferior frontal gyrus (IFG), parahippocampal gyrus (PHG), superior temporal cortex (STG), and dorsolateral prefrontal cortex (DLPFC). (B) Comparison of RNA sequencing-based enrichment between known AD gene sets and GeneEMBED candidates. Stars indicates the number of brain regions with significant enrichment in gene set. Violin plot shows the distribution of expected number of enriched brain regions when using random gene sets. (C) Among the 143 high-confidence genes, a significant number (22, $p=0.0247$) showed differential expression in both bulk tissue from various brain regions and in single cell sequencing of neuronal cell types.”

- Figure 3, all axis labels should be described in the legend.

Response: We thank the reviewer for noting this point. We have adjusted the figure legend, which now reads:

“Figure 3. GeneEMBED candidates are significantly related to curated sets of AD genes. (A) Receiver Operator Characteristic curves are shown for Disc. VISEA for network diffusion to CTD and ClinVar AD gene sets. To determine significance of observed AUC, random gene sets of the same size are generated 100 times and analyzed through nDiffusion to create a random distribution of AUCs. Reported z-scores are calculated relative to these backgrounds. Y-axis of the ROC plots are true positive rates (TPR) and x-axis is false positive rate (FPR). Similarly, y-axis of the z-score distribution is probability density and x-axis is the AUROC score of random gene sets. Analogous plots are shown for (B) Ext VISEA, (C) Disc VISPPH2, and (D) Ext VISPPH2”

Figures and tables generated from the thoughtful comments of the reviewers are shown below.

Fig S2. GeneEMBED candidates are consistently identified across various cohorts, networks, and VIS systems. Hypergeometric overlap tests were done on every pairwise combination of cohort-network-VIS experiments. Among 66 independent pairwise tests, only 11 did not demonstrate statistically significant hypergeometric p-values ($p < 0.05$, $\log(p) < -2.99$).

Fig S7. GeneEMBED is robust to false negative and false positive edges. (A) Edges were synthetically and randomly deleted from the Brain network to test sensitivity of GeneEMBED to false negative edges. In blue are plots of precision and recall of GeneEMBED identified genes at various levels of randomly deleted edges. In red are plots of precision and recall of GeneEMBED identified genes when randomly deleted edges are targeted for known (previously identified) genes. (B) Edges were synthetically and randomly added to the Brain network to test sensitivity of GeneEMBED to false positive edges. The plot shows precision and recall of GeneEMBED identified genes at various levels of synthetically added edges. X-axis of ‘% Edges Added’ is relative to the original network size, e.g. at 100%, ~48k edges are randomly added.

Human GeneID	Drosophila Homolog	DIOPT Score (max 15)	Allele Class	Allele type	Specific alleles
ABL1	Abl	9	LOF (loss of function)	Amorphic (W559term)	Abl[2]/TM68, Tst[1]
ABL1	Abl	9	OE (Over expression)	Myc tagged cDNA inducible	w[*], P[w{mC}=UAS-Abl.Myc]attP40
BAG3	stv	9	LOF	Insertion of P-element disrupting intron 2 and an alternative promoter	P[ry{+7.2}-P2]stv(00543) ry[506]/TM3, ry[RK] Sb[1] Ser[1]
Blm	Blm	10	LOF	Imprecise excision of the P[EGy2]EV03745.	w[118], Blm[N1]/TM3, Sb[1]

Supplementary Table 12 (truncated version here): Drosophila homologues of high confidence candidate genes tested in in vivo fly models.

Gene Set (GS)	Original Experiment (n=69)		Full Embedding distance (n = 82)		Max PS (n = 73)		PS Threshold (PS < 0.7)	
	Ovlp	Pval	Ovlp	Pval	Ovlp	Pval	Ovlp	Pval
GWAS Meta 1 (n = 25)	1	0.091	2	0.006	1	0.101	1	0.1
GWAS Meta 2 (n = 38)	1	0.142	2	0.017	1	0.16	1	0.158
Comp. Tox. Database (n = 103)	2	0.047	2	0.073	2	0.057	2	0.057
ClinVar (n = 21)	1	0.081	1	0.1	1	0.088	1	0.087
DisGeNet (n = 208)	5	1.34E-03	7	7.40E-05	5	1.97E-03	5	1.85E-03

Supplementary Table 13: Hypergeometric overlaps between reference gene sets and candidate genes generated from the (i) original GeneEMBED framework, (ii) distances computed on full dimensional embeddings, (iii) edge weighting scheme of max(PS), (iv) edge weighting scheme of PS thresholding.

Gene Set (GS)	Original Experiment (n=69)		Full Embedding distance (n = 82)		Max PS (n = 73)		PS Threshold (PS < 0.7) (n=72)	
	AUC	Z-score	AUC	Z-score	AUC	Z-score	AUC	Z-score
GWAS Meta 1 (n = 25)	0.74	3.71	0.73	2.82	0.73	2.81	0.75	3.47
GWAS Meta 2 (n = 38)	0.63	2.54	0.64	2.05	0.63	1.8	0.65	2.73
Comp. Tox. Database (n = 103)	0.76	2.03	0.78	2.74	0.75	1.27	0.76	1.64
ClinVar (n = 21)	0.78	2.3	0.78	1.96	0.76	2.75	0.77	3.53
DisGeNet (n = 208)	0.69	3.26	0.67	0.97	0.72	1.51	0.7	2.27

Supplementary Table 14: nDiffusion analyses between reference gene sets and candidate genes generated from the (i) original GeneEMBED framework, (ii) distances computed on full dimensional embeddings, (iii) edge weighting scheme of max(PS), (iv) edge weighting scheme of PS thresholding.

	Original Experiment (n = 69)	Full Embedding Distances (n = 84)	Max PS (n = 73)	PS Threshold (PS < 0.7) (n = 72)
Num. of Regions Enriched	2	0	1	2
p-value	0.012		0.076	0.0014

Supplementary Table 15: Enrichment of genes generated by (i) original GeneEMBED framework, (ii) distances computed on full dimensional embeddings, (iii) edge weighting scheme of max(PS), (iv) edge weighting scheme of PS thresholding for differentially expressed genes in various brain regions from post-mortem AD brains.

Gene Set (GS)	STRING		HINT		Brain		HuRI	
	Ovlp	Pval	Ovlp	Pval	Ovlp	Pval	Ovlp	Pval
GWAS Meta 1 (n = 25)	1	0.091	2	0.005	1	0.051	0	1
GWAS Meta 2 (n = 38)	1	0.142	2	0.015	1	0.057	0	1
Comp. Tox. Database (n = 103)	2	0.047	4	0.001	2	0.024	1	0.34
ClinVar (n = 21)	1	0.081	1	0.11	1	0.068	0	1
DisGeNet (n = 208)	5	1.34E-03	3	0.11	2	0.096	1	0.58

Supplementary Table 16: Hypergeometric overlap between reference gene sets and genes identified by GeneEMBED using (i) STRING network, (ii) HINT network, (iii) Brain specific PPI network, and (iv) unbiased HuRI network.

Gene Set (GS)	STRING		HINT		Brain		HuRI	
	AUC	Z-score	AUC	Z-score	AUC	Z-score	AUC	Z-score
GWAS Meta 1 (n = 25)	0.74	3.71	0.68	1.86	0.71	2.69	0.51	-1.59
GWAS Meta 2 (n = 38)	0.63	2.54	0.62	2.37	0.63	2.46	0.54	-0.12
Comp. Tox. Database (n = 103)	0.76	2.03	0.77	4.34	0.72	2.11	0.68	0.48
ClinVar (n = 21)	0.78	2.3	0.76	3.34	0.78	3.91	0.66	0.76
DisGeNet (n = 208)	0.69	3.26	0.7	5.77	0.66	2.15	0.61	-0.41

Supplementary Table 17: nDiffusion analyses between reference gene sets and genes identified by GeneEMBED using (i) STRING network, (ii) HINT network, (iii) Brain specific PPI network, and (iv) unbiased HuRI network.

	STRING	HINT	Brain	HuRI
Num. of Regions Enriched	2	1	0	1
p-value	0.012	0.048	1	0.06

Supplementary Table 18: Enrichment of genes generated using GeneEMBED on (i) STRING, (ii) HINT network, (iii) Brain specific PPI network, and (iv) unbiased HuRI network for differentially expressed genes in various brain regions from post-mortem AD brains.

EA Analyses	CTD	GWAS Meta 1	GWAS Meta 2	DGN	ClinVar
observed	0.003991	0.022988998	0.0356475	0.015473	0.019344
random 1	0.116209	1	1	0.221469	1
random 2	0.128265	1	1	0.242841	1
random 3	1	1	1	1	1
random 4	0.151891	1	1	0.283848	1
random 5	0.128265	1	1	0.242841	1

Supplementary Table 19 (truncated version here): Hypergeometric overlap between observed reference gene sets and genes identified overlapping between cohorts when case and control labels are correctly assigned (observed) and when they are randomized (random) using VIS_{EA}. Only the first 6 rows are shown here for brevity, for the full table please see supplementary table file included separately.

PPh2 Analyses	CTD	GWAS Meta 1	GWAS Meta 2	DGN	ClinVar
observed	1	1	1	1	1
random 1	1	1	1	0.368211	1
random 2	1	1	1	0.332012	1
random 3	1	1	1	1	1
random 4	1	1	1	1	1
random 5	1	1	1	0.303515	1

Supplementary Table 20 (truncated version here): Hypergeometric overlap between observed reference gene sets and genes identified overlapping between cohorts when case and control labels are correctly assigned (observed) and when they are randomized (random) using VIS_{PPh2}. Only the first 6 rows are shown here for brevity, for the full table please see supplementary table file included separately.

Gene	STRING-Disc-EA	STRING-Disc-PPh	STRING-Ext-EA	STRING-Ext-PPh	HINT-Disc-EA	HINT-Disc-PPh	HINT-Ext-EA	HINT-Ext-PPh	BRAIN-Disc-EA	BRAIN-Disc-PPh	BRAIN-Ext-EA	BRAIN-Ext-PPh	In Known AD gene	Dysregulated in AD brains	Neuro. Phen. in Mouse KO	Modifier of AD in fly	Total Score
APCE																	12
MAP3B																	10
TP53																	6
MAPT																	6

Supplementary Table 21 (truncated version here): Summary table of the union of all genes identified across all twelve GeneEMBED analyses. Columns indicate in which analyses each gene was identified as well as other sources of evidence supporting the gene's association to AD. Red colored cells indicate the gene was identified in a particular analysis or had positive findings in the corresponding analysis. White colors indicate the opposite. The final column gives the sum of the number of red cells of each gene. Only the first 4 rows are shown here for brevity, for the full table please see supplementary table file included separately.

Referees' report, second round of review

Reviewer #1: The authors have done a superb job in addressing our comments, and have presented the revisions in a very clear way. We see that, unfortunately, GeneEMBED is not able to identify significant AD-genes starting from unbiased networks, but we agree with the authors that it could be due to some limitations of these networks. Overall, we congratulate them on the work, and are now happy to recommend the publication of the m/s in CELL GENOMICS.

Reviewer #2: The authors have addressed the issues/questions I've raised and they've given more information about the performance of the approach they developed. I am happy with the publication of the updated version.

Reviewer #3: The authors extensively addressed the reviewers comments and revised the manuscript accordingly.
I am happy to support its publication as presented.

Authors' response to the second round of review

Response to the reviewers:

We thank the three reviewers for their thorough and thoughtful inputs throughout the review process. We are grateful for the many overall positive comments from all three reviewers on the quality, novelty, and impact of our study. No further comments or changes are requested from the reviewers at this stage.

For clarity:

Comments from the reviewers are in Times italics, black.

Our responses to the reviewers are in Arial, black, indented

Reviewer #1 wrote:

The authors have done a superb job in addressing our comments, and have presented the revisions in a very clear way. We see that, unfortunately, GeneEMBED is not able to identify significant AD-genes starting from unbiased networks, but we agree with the authors that it could be due to some limitations of these networks. Overall, we congratulate them on the work, and are now happy to recommend the publication of the m/s in CELL GENOMICS.

We thank R1 for these positive comments on the writing and presentation of our revisions.

Reviewer #2 wrote:

The authors have addressed the issues/questions I've raised and they've given more information about the performance of the approach they developed. I am happy with the publication of the updated version. We thank R2 for their positive comments regarding the information we provided about the approach presented in this study.

Reviewer #3 wrote:

The authors extensively addressed the reviewers comments and revised the manuscript accordingly. I am happy to support its publication as presented.

We thank R3 for their positive comments regarding the revisions we have presented.